# Neural excursions from manifold structure explain patterns of learning during human sensorimotor adaptation

Corson Areshenkoff[1,2]*, Daniel J Gale[1], Dominic Standage[3], Joseph Y Nashed[1], J Randall Flanagan[1,2], Jason P Gallivan[1,2,4]

[1]Centre for Neuroscience Studies, Queen's University, Kingston, Canada; [2]Department of Psychology, Queen's University, Kingston, Canada; [3]School of Psychology, Centre for Computational Neuroscience and Cognitive Robotics, University of Birmingham, Birmingham, United Kingdom; [4]Department of Biomedical and Molecular Sciences, Queen's University, Kingston, Canada

**Abstract** Humans vary greatly in their motor learning abilities, yet little is known about the neural mechanisms that underlie this variability. Recent neuroimaging and electrophysiological studies demonstrate that large-scale neural dynamics inhabit a low-dimensional subspace or manifold, and that learning is constrained by this intrinsic manifold architecture. Here, we asked, using functional MRI, whether subject-level differences in neural excursion from manifold structure can explain differences in learning across participants. We had subjects perform a sensorimotor adaptation task in the MRI scanner on 2 consecutive days, allowing us to assess their learning performance across days, as well as continuously measure brain activity. We find that the overall neural excursion from manifold activity in both cognitive and sensorimotor brain networks is associated with differences in subjects' patterns of learning and relearning across days. These findings suggest that off-manifold activity provides an index of the relative engagement of different neural systems during learning, and that subject differences in patterns of learning and relearning are related to reconfiguration processes occurring in cognitive and sensorimotor networks.

**\*For correspondence:**
areshenk@protonmail.com

**Competing interest:** The authors declare that no competing interests exist.

## Editor's evaluation

This manuscript describes a fascinating experiment looking at gross network dynamics across cognitive and motor circuits and across different stages of learning, during an adaptive visuo-motor learning experiment in the MRI environment. The finding of reliable "excursions" from low-dimensional network states that are associated with learning, primarily in cognitive networks, and that this excursion metric is a reliable indicator of differences in learning has strong implications for our understanding of the way macroscopic brain networks learn new skills.

## Introduction

Effective motor behavior relies on the central nervous system's ability to adapt to the environment by learning new mappings between motor commands and sensory outcomes. For example, an inexperienced youth hockey player, when walking from the change-room onto the ice surface, has to learn to adapt their stride in response to the drastic change in surface properties. This sensorimotor adaptation is known to rely partly on implicit learning processes, driven by sensory prediction errors reflecting the difference between the expected and sensed consequences of movement (**Wolpert et al., 1995**; **Wolpert and Ghahramani, 2000**). In addition, there is evidence that explicit (declarative)

learning processes, such as cognitive strategies, also help guide behavioral adjustments, and that these operate synergistically with implicit processes (see *Krakauer and Mazzoni, 2011*; *McDougle et al., 2016*; *Wolpert et al., 2011*; *Miyamoto et al., 2020*). Recent behavioral and computational work (*McDougle et al., 2015*) has suggested that these two processes operate on different timescales, with implicit and explicit learning leading to slow and fast adaptation and de-adaptation, respectively.

Although the neural systems supporting these two learning processes are not perfectly understood, there is considerable evidence that implicit adaptation relies, in part, on sensory prediction errors computed in the cerebellum (*Izawa et al., 2012*; *Taylor et al., 2014*), which are used to drive adaptation in frontal and parietal sensorimotor regions in the neocortex (*McDougle et al., 2015*). Explicit learning, by contrast, has been linked to higher-level working memory and inhibitory control processes (*Keisler and Shadmehr, 2010*; *Anguera et al., 2010*; *Christou et al., 2016*; *Drummond et al., 2015*), and thus has been associated with a wide array of higher-order association areas across the neocortex (*Anguera et al., 2007*; *Dayan and Cohen, 2011*; *Doyon et al., 2009*; *Hardwick et al., 2013*). There is also evidence that explicit learning is responsible for 'savings', or faster relearning upon reexposure to the task (*Haith et al., 2015*; *Morehead et al., 2015*). Moreover, emerging behavioral evidence indicates that individual differences in rates of adaptation and savings is largely driven by subject-level differences in the contribution of explicit processes to learning (*de Brouwer et al., 2018*; *Fernandez-Ruiz et al., 2011*). While the neural bases that underlie these subject-level differences in learning performance are likewise poorly understood, one hypothesis that follows from this prior work is that fast and slow learners are likely to differ in the relative degree to which they recruit brain networks associated with these distinct learning systems. These differences in the engagement of sensorimotor networks on the one hand, and higher-order cognitive regions on the other, may then reflect the differential contribution of explicit cognitive processes to their learning performance.

At the neural level, learning requires that the brain generates new patterns of activity. These activity patterns are presumably constrained by the underlying neural circuitry and shaped by the information processing functions of single neurons or individual brain regions. This architecture results in correlations between the activity of neurons (and between brain regions), resulting in patterns of activation that primarily occupy a low-dimensional subspace or manifold (*Gallego et al., 2017*; *Sadtler et al., 2014*; *Shenoy et al., 2013*). In neurophysiological studies, this manifold structure has been observed both at the level of whole-brain functional networks (via functional MRI [fMRI]; *Shine et al., 2019*) and at the level of local neural populations (*Sadtler et al., 2014*; *Gallego et al., 2017*). Recent work has suggested that some types of learning are associated with a reconfiguration of this low-dimensional structure, allowing the brain to generate new patterns of off-manifold activity (*Shine et al., 2019*; *Oby et al., 2019*). Here, we wondered whether differences in the rate of learning and relearning during sensorimotor adaptation could be explained by differences in the generation of off-manifold activity in cognitive and sensorimotor networks; and, if so, whether differences in cognitive and sensorimotor excursion are associated with distinct patterns of subject performance.

We used fMRI to study changes in intrinsic manifold structure during sensorimotor adaptation on 2 consecutive days. Analysis of subject behavior revealed three broad clusters of learners: Subjects who adapted and dea-dapted rapidly on both days; subject who adapted and de-adapted slowly on both days; and a third group of subjects who learned slowly on day 1, but de-adapted rapidly and relearned rapidly on day 2. We then examined excursions from intrinsic manifold structure in cognitive and sensorimotor networks (i.e., off-manifold activity) during learning, and found that the degree of excursion in these networks is related to differences in these group's patterns of learning and relearning across days. Finally, to further characterize these changes, we analyzed functional network structure across learning epochs by embedding networks from each subject and learning phase of the task into a common space. Together, our findings suggest that excursion from intrinsic manifold structure provides an index of the relative engagement of distributed brain networks during learning, and that fast versus slow patterns of learning across days are associated with different reconfiguration processes of cognitive and sensorimotor systems.

## Results

To study adaptation learning, subjects (*N* = 32) performed a visuomotor rotation (VMR) task on 2 consecutive days, requiring them to launch a cursor to intercept a stationary target by applying a brief, isometric directional force pulse on an MRI-compatible force sensor. The task structure on each

day was identical: After a series of baseline trials (baseline epoch), the mapping between force and cursor motion was rotated clockwise by 45°, such that subjects had to learn to counteract the rotation in order to successfully hit the target. Following this learning epoch, the rotation was removed (washout epoch) to allow subjects to de-adapt before relearning on the second day (see *Figure 1a* for trial structure).

Here, we took two approaches to study neural changes associated with learning. First, we examined learning-induced changes in cognitive and sensorimotor network structure (i.e., following the onset of the VMR) using a neural manifold approach inspired by the works of *Sadtler et al., 2014*, and *Shine et al., 2019*. This allowed us to study the degree to which activity in each network during learning deviates from its intrinsic manifold structure, assessed during a prior resting-state scan — a measure which we will hereby refer to as *excursion* (*Figure 1b*). We then related the degree of this neural excursion in the cognitive and sensorimotor brain networks to subjects' patterns of learning performance within and across days. Second, we examined changes in functional network structure during learning by estimating covariance networks from the cognitive and sensorimotor regions during the baseline, early, and late learning epochs (*Figure 1c*). We then derive a low-dimensional embedding of these networks, which allows us to directly characterize subject differences in network structure across each of these epochs of the task, as well as across days (*Figure 1d*).

## Clustering of behavior reveals qualitatively distinct groups of learners across days

Mean error curves across subjects for each day are shown in *Figure 2a* (top), confirming that subjects successfully reduced their error over the course of the learning epoch. Since a full characterization of subject performance would require summarizing rates of learning, unlearning, and the total adaptation on both days (along with other features of performance, e.g., reaction times, which have been linked to distinct cognitive or motor processes), here we sought an interpretable, low-dimensional summary amenable to simple statistical analysis. This approach was motivated by our observation that standard summary measures generally failed to capture overall performance patterns across both days (see also *Standage et al., 2020*). For example, 'savings', the difference in early learning performance across days, failed to distinguish between subjects who learned the rotation rapidly on both days versus those who learned and relearned slowly. Similarly, early learning did not distinguish between subjects who adapted and de-adapted slowly on the first day versus those who learned slowly but showed performance features similar to fast learners by the end of the first day. In order to properly profile these differences in performance, we clustered individual subjects by early error on days 1 and 2 (defined as the mean error over the first 4 eight-trial blocks of the rotation phase), along with savings (the difference in early error across days). This clustering revealed three distinct subgroups of learners (*Figure 2b*), which corresponded to subjects who exhibited fast learning on both days (FF, *N* = 16), slow learning on both days (SS, *N* = 9), and slow learning on day 1 but fast learning on day 2 (SF, *N* = 7; *Figure 2c*). We evaluated this cluster solution using the procedure described by *Kimes et al., 2017*. For numbers of clusters between 2 and 8, we computed the ratio of within/between cluster sums of squares and compared this to a null model of a multivariate normal distribution with covariance equal to the observed covariance between variables. Based on these simulations, we judged that a three cluster solution best explained the data (*Figure 2b*, bottom). This clustering solution had the dual benefit of both capturing the dominant patterns of variability in subject performance, and of producing a simple performance measure that could be compared to each subject's neural data.

The rapid adaptation and de-adaptation shown by the FF group on both days, and by the SF group on day 2, are considered signatures of explicit learning (*Benson et al., 2011*; *de Brouwer et al., 2018*). In the context of the VMR task, this includes the use of a re-aiming strategy, in which a subject aims so as to counteract the rotation applied to the cursor (*Taylor et al., 2014*). One possible interpretation of the SF group's performance is that they adopt an explicit strategy early during day 2, after learning largely implicitly on day 1. Another alternative is that this group learns explicitly, even on day 1, but employs an explicit strategy either poorly or too late to reduce their early error. This second possibility is supported by similarity between the FF and SF groups later on day 1 (*Figure 2d*): both groups show comparable performance during late learning and washout (*Figure 2c*). The FF and SF groups also display relatively long reaction times compared to the SS group (*Figure 2d*), which have been associated with explicit strategy use (e.g., *Fernandez-Ruiz et al., 2011*; *Haith et al., 2015*).

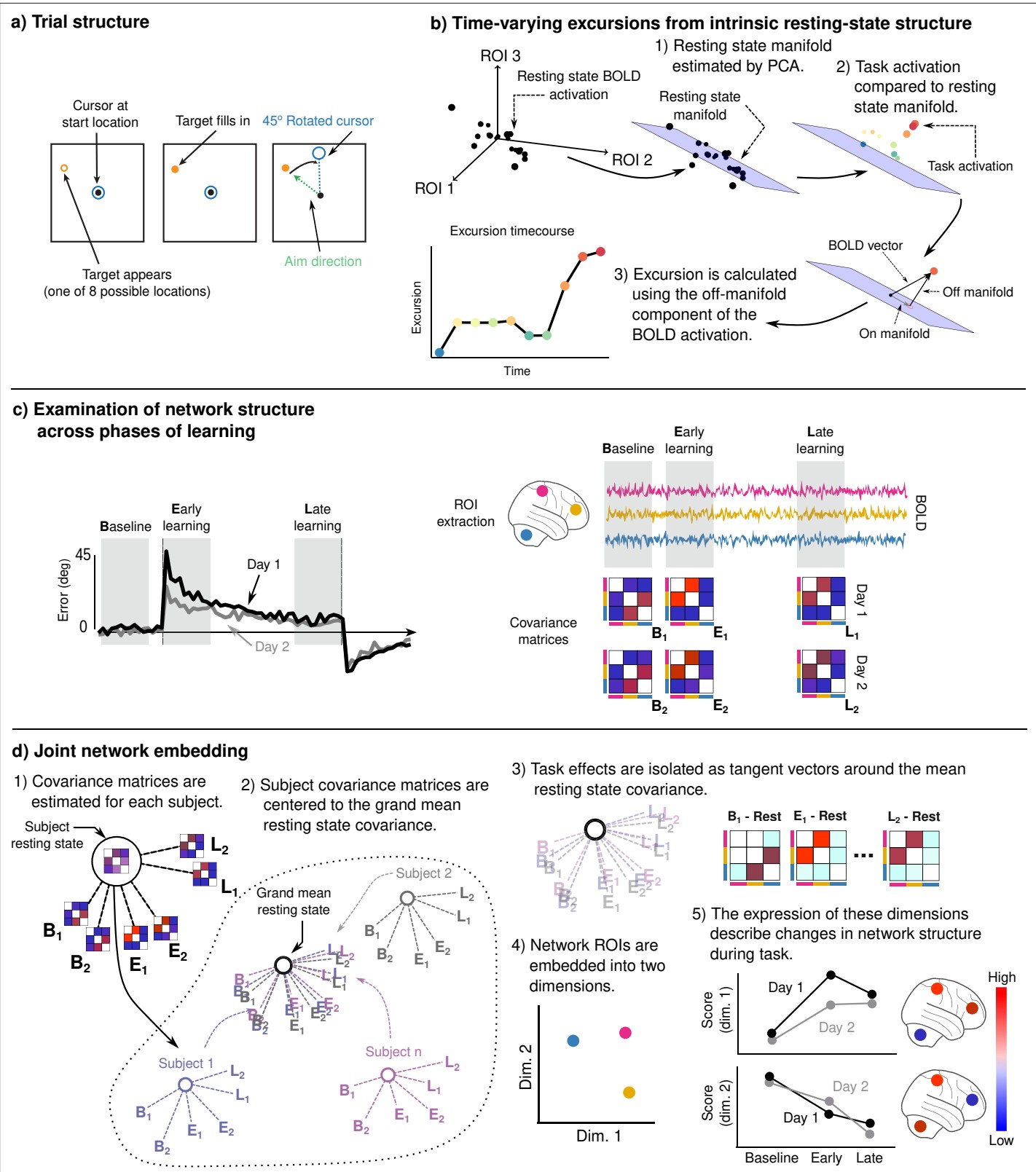

**Figure 1.** Adaptation task and analysis approach for the neural data. (**a**) Visuomotor rotation (VMR) trial structure. Subjects used a force-sensitive joystick to move a cursor to a cued target location. On rotation trials, the mapping between aim and cursor direction was rotated clockwise by 45°. (**b**) We studied the degree to which learning is associated with the generation of novel patterns of neural activity by estimating resting-state manifolds for each subject, and then calculating the magnitude of the off-manifold component of the BOLD signal at each imaging volume during task. This

*Figure 1 continued on next page*

*Figure 1 continued*

measure, which we call *excursion*, provides a moment-by-moment index of functional reconfiguration of a brain network over the course of learning. (**c**) We studied functional network structure during baseline, early, and late learning on each day (mean subject learning curves shown on left over these epochs) by estimating covariance matrices from the mean BOLD signals extracted from cognitive and motor regions of interest (ROIs). (**d**) We isolated learning-related changes in functional network structure by centering subjects with respect to their resting-state network structure. We then jointly embedded networks estimated during each learning epoch into a common space in order to characterize changes in network structure over the time course of learning.

As explicit learning is believed to be highly cognitive, we may expect that the rapid learning exhibited by the FF group would be associated with the greater recruitment of cognitive brain regions outside of traditional sensorimotor networks, commonly associated with adaptation to sensorimotor errors. Moreover, if the late performance of the SF group on day 1 is indeed reflective of explicit strategy use, we might expect to see similar patterns of neural recruitment between the FF and SF groups.

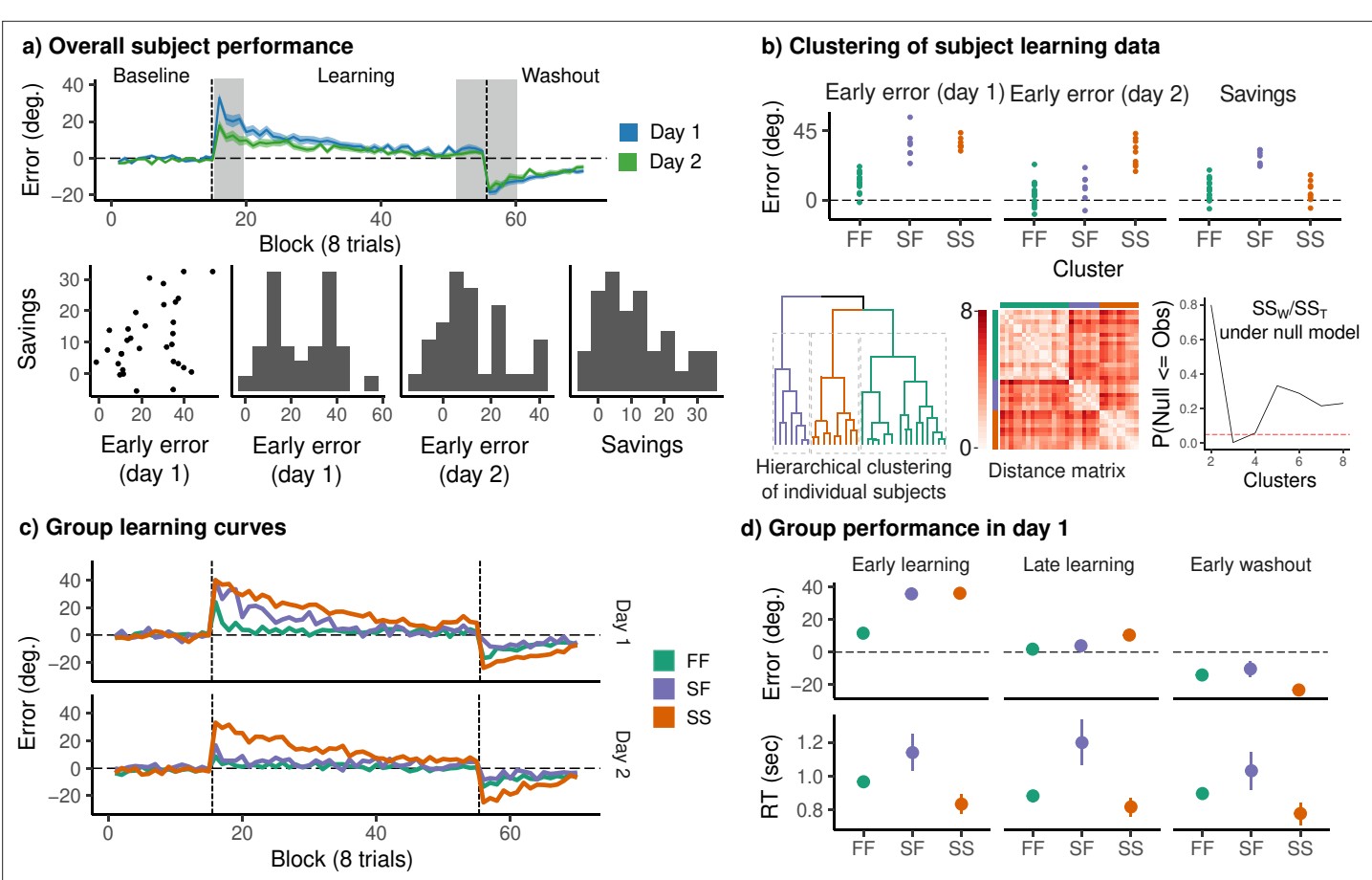

**Figure 2.** Clustering reveals three distinct groups of learners across days. (**a**) (Top) Mean angle error during baseline, rotation, and washout on both days. Gray bands denote early and late rotation periods, and early washout. Colored bands denote standard errors. (Bottom) Early error on each day (mean error during the early learning window), as well as savings. (**b**) (Top) Variables used for clustering. (Bottom, from left) Dendrogram obtained by hierarchical complete linkage clustering on the $\ell_1$ distance for the above variables for each subject. Distance matrix displaying the distance between individual subjects' clustering variables. The cluster solution was evaluated using the procedure described by *Kimes et al., 2017*. The cluster solution was compared to a null model in which the data were sampled from a single multivariate normal distribution with identical covariance to the observed data. The observed data were compared with 1000 bootstrap samples for a number of clusters ranging from 2 to 8. The three cluster solution was chosen to form the basis for our analysis. (**c**) Mean error curves for each behavioral group. (**d**) Mean error and reaction time during early learning, late learning, and early washout on day 1.

## FF and SF groups show greater excursion from intrinsic manifold structure at the onset of learning

In order to quantify the engagement of cognitive and sensorimotor systems during learning (and, in particular, at the onset of the VMR), we developed a novel measure, called *excursion*, quantifying the degree to which the functional organization of these networks deviates from the structure present during rest. Our approach is motivated by recent neural recording work (*Sadtler et al., 2014*) investigating whether deviations from low-dimensional manifold structure are related to the capacity for learning, as well as fMRI work investigating how these deviations relate to task performance (*Shine et al., 2019*).

We first defined cognitive and sensorimotor networks using the whole-brain parcellation proposed by *Seitzman et al., 2020*, comprising 300 regions of interest (ROIs) across the cortex, basal ganglia, and cerebellum, and assigned functional classifications according to the parcellation of *Power et al., 2011*. Given that nearly every part of the brain can be said to be involved in 'cognition' in some capacity, we simplified our division of motor versus cognitive networks such that it distinguished between higher-order association areas implicated in attention and cognitive control (frontoparietal, dorsal attention, and ventral attention networks, FP, DA and VA, respectively; 62 ROIs), and regions primarily involved in the control and generation of movement (somatomotor dorsal and somatomotor lateral networks, SMD and SML, respectively; 51 ROIs). We note that our cognitive network comprises a large part of what has been characterized as the 'multiple-demand system' (*Duncan, 2010*) in frontal and parietal cortex, which has been associated with diverse cognitive processes and the implementation of strategies. The DA and VA networks in particular have been implicated in motor inhibition (*Hsu et al., 2020*) and in the reorienting of spatial attention (*Simpson et al., 2011*; *Corbetta and Shulman, 2002*), both processes of which are involved in explicit strategy use during VMR learning (*de Brouwer et al., 2018*). A more exploratory analysis, comprising the individual functional networks making up the Seitzman parcellation, is presented in *Figure 3—figure supplement 1*.

We used principal component analysis (PCA) to estimate a low-dimensional subspace capturing the dominant patterns of BOLD signal covariance in each network during each subject's resting-state scan. We then examined excursions from this intrinsic structure during the VMR task, defined as the proportion of the BOLD signal activation vector which lies off of the subject's intrinsic manifold. This

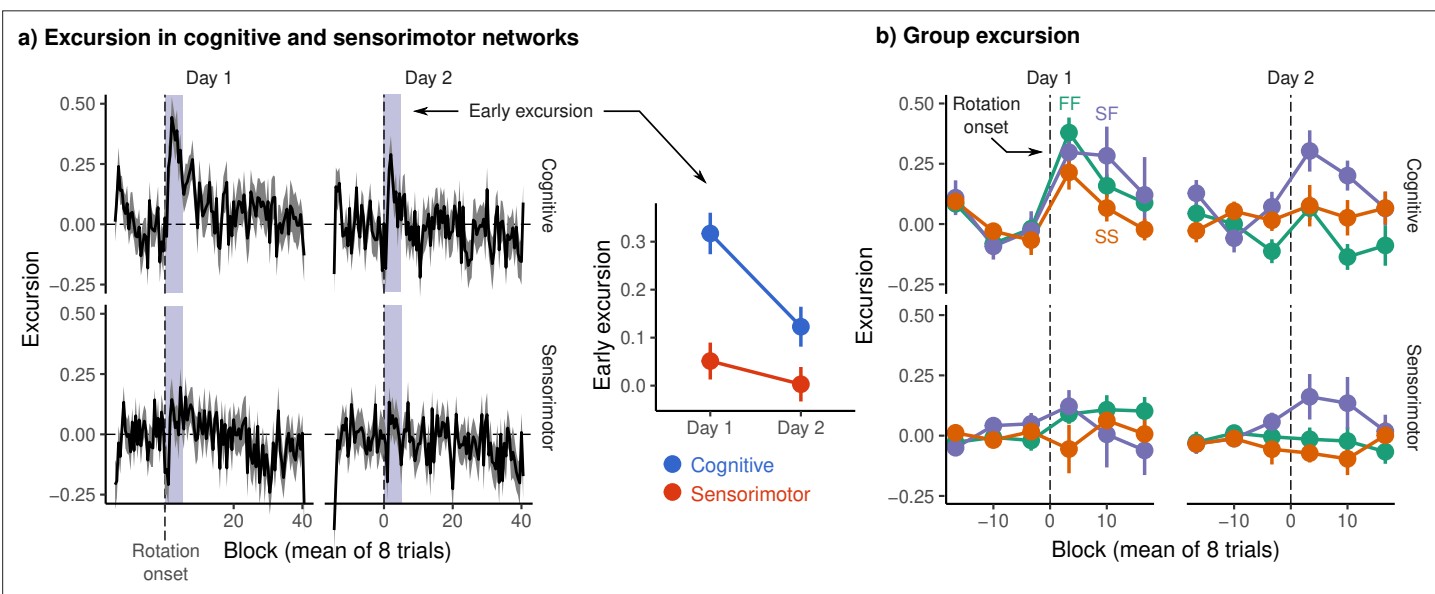

**Figure 3.** Excursion from resting manifold structure. (**a**) (Left) Mean excursion in cognitive and sensorimotor networks on each day. In order to isolate the effect of rotation, subject excursion curves were standardized with respect to baseline. Vertical dashed line denotes the onset of rotation. (Right) Early excursion, defined as the mean excursion during the first 4 eight-trial blocks (shaded region in left panel), analogous to our definition of early error. (**b**) Mean excursion for each behavioral group. Error bars represent standard errors.

The online version of this article includes the following figure supplement(s) for figure 3:

**Figure supplement 1.** Excursion for individual Seitzman networks.

measure provides an index of the degree to which activation within the cognitive and sensorimotor networks deviate from the structure present during rest. For each subject, we computed this neural excursion at each timepoint (imaging volume) in the cognitive and sensorimotor networks on each day. We then baseline-centered these excursion curves in order to isolate changes in excursion during the learning phase (see Materials and methods for a more detailed description).

*Figure 3a* shows the mean excursion across subjects in the cognitive and sensorimotor networks on each day. Notably, we observed a sharp peak in excursion in the cognitive network immediately after the onset of rotation, which is most pronounced on day 1, but also prominent on day 2. We defined *early excursion* to be the mean excursion during the first 4 eight-trials blocks – equivalent to our behavioral definition of early error – and compared excursion across networks and days using a 2 × 2 factorial analysis of variance (ANOVA) with day and network as within-subjects effects. After correcting for multiple comparisons (*Benjamini and Hochberg, 1995*), we observed significant main effects of day ($F_{1,29} = 10.197$, p = .002, q = .003) and of network ($F_{1,29} = 25.766$, p < .001, q < .001), but no day by network interaction ($F_{1,29} = 3.691$, p = .06, q = .06). *Figure 3b* shows the post-rotation peak broken down by behavioral group, revealing what appears to be greater peak magnitude in the FF and SF groups (compared to SS) in the cognitive network on day 1, and a greater peak in the SF group on day 2, where these subjects succeed in reducing their early error in a similar fashion to FF subjects.

Due to the complexity of these data – which comprise excursion at each timepoint, in each network, and across 2 days – we sought an interpretable, low-dimensional representation of the patterns of

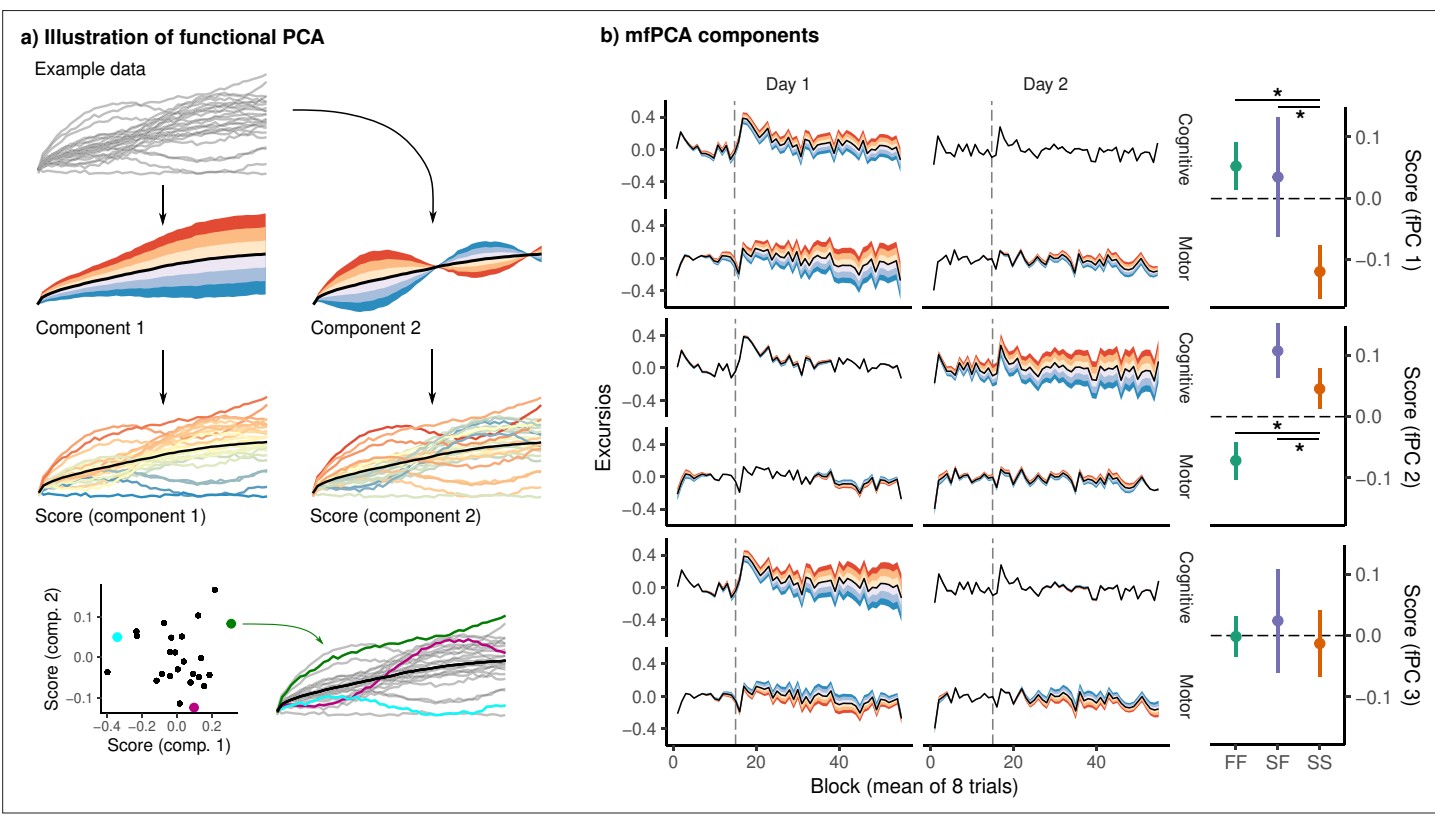

**Figure 4.** Functional principal component analysis (fPCA) of network excursion data. (**a**) An example illustration of fPCA. Given a set of curves (top), fPCA extracts components describing the dominant patterns of variability in functional data (second row). In this example, the first component encodes whether the curve is higher or lower than average, while the second encodes a sinusoidal pattern in which the curve begins higher or lower than average, and then reverses direction. The score associated with a component then describes the degree to which that component is expressed in a particular observation (third row). These scores provide a low-dimensional representation of the overall shape of the curve. For example, in the bottom row, we see three example curves, and their corresponding scores on the two components. (**b**) Functional principal components extracted from subject excursion curves. Each panel corresponds to a single component, representing a pattern of variability in excursion across each network and day. The left panel illustrates each component as a deviation from the mean excursion (solid black line). Red (resp. blue) bands denote the effect of positive (resp. negative) scores. The right panel shows the mean score in each group. Error bars denote standard errors, while stars denote significant differences for pairwise Conover-Iman tests with Benjamini-Hochberg correction for multiple comparisons within each component.

excursion across days and across networks in order to reduce the total number of possible comparisons. To this end, we used multivariate functional principal component analysis (mfPCA) to derive a low-dimensional summary of the main patterns of excursion across individual subjects during learning. Just as standard PCA finds a set of components capturing the dominant patterns of variability of data lying in ordinary Euclidean space $\mathbb{R}^n$, fPCA attempts to find a basis of component functions capturing the dominant patterns of variability in functional data (see *Figure 4a* for an example illustration). As each subject contributes four excursion curves – one for each network and each day – we used mfPCA (*Happ and Greven, 2018*) to summarize the dominant patterns of excursion across days and networks. Based on variance explained by each component, we retained three components for analysis, which together describe 77% of the variance across all excursion curves. Each of these components describes a characteristic pattern of excursion, represented as a deviation from the overall mean. Note that the identification of these components was done over all subjects, and thus is agnostic to group membership (i.e., FF versus SS versus SF subjects).

The first of these components is shown in *Figure 4b*. In the left panel, the black lines denote the mean excursion across all subjects, while the colored bands denote the modes of variation encoded by the component, with red (resp. blue) denoting the effects of positive (resp. negative) scores. The right panel shows mean scores on this component for each of the three groups. In this case, the first component appears to describe overall excursion on day 1, with positive scores denoting greater excursion than average in both networks. Positive scores on the second component appear to encode greater excursion in the cognitive network on day 2. Finally, component 3 appears to encode cognitive excursion on day 1, with a slight decrease in motor excursion on both days.

Notably, we observed significant group differences in the expression of the first (Kruskal-Wallis, $\chi_2^2 = 6.03$, p = .04) and the second components ($\chi_2^2 = 10.16$, p = .006). For each of these components, we conducted pairwise Conover-Iman post hoc tests (*Conover and Iman, 1979*) between groups, correcting for multiple comparisons (*Benjamini and Hochberg, 1995*). For component 1, scores were significantly higher in the FF group than in the SS group ($z = 2.62$, p = .02), as well as higher in the SF group than in the SS group ($z = 2.25$, p = .02). For component 2, scores were significantly higher in the SF group than in the FF group ($z = 3.29$, p = .004), as well as higher in the SS group than in the FF group ($z = 2.35$, p = .02). Finally, for component 3, we found no significant effect of group.

In summary, interpreting both the results of the mfPCA and the mean excursion shown in *Figure 3a*, both the FF and SF groups show a large degree of excursion in the cognitive network on the first day, with the SF group likewise showing a spike in excursion on day 2. These results are consistent with the idea that the rapid learning exhibited by the FF group involves the recruitment of cognitive brain regions outside the sensorimotor cortices. The fact that the SF group expresses a similar excursion profile on day 1 lends evidence to the interpretation that the SF does indeed engage cognitive learning processes on day 1, as reflected in their late performance, but merely fails to learn a strategy early enough to bring down their early error.

## FF and SF groups show similar changes in cognitive network structure during learning

We now turn our attention to the interpretation of the patterns of neural excursion described in the previous section. To help characterize these neural changes, and allow a direct comparison of functional network structure across different phases of the task, we applied an approach based on the simultaneous embedding of functional networks estimated during baseline, early, and late learning.

For each of these networks, we estimated covariance matrices from the subject's resting-state scan (177 TRs), and equivalent length periods on each day during (1) the beginning of baseline, (2) immediately following the onset of rotation (early learning), and (3) at the end of the rotation block (late learning; see *Figure 1b* for an illustration).

## Covariance centering reveals task-related changes in network structure

Previous work (e.g., *Gratton et al., 2018*) has found that subjects exhibit strong individual differences in functional connectivity, which are likely to obscure learning- or task-related changes in network structure. To remove these subject-specific differences prior to analysis, we centered each subject's task-related covariance matrices with respect to their resting-state covariance matrix using a procedure modelled after the approach of *Zhao et al., 2018* (a full description is given in the Materials

and methods). To illustrate the effect of this centering procedure – and why it is critical for investigating any learning-related effects here – we used uniform manifold approximation (UMAP; *McInnes et al., 2018*) to project the individual covariance matrices into two dimensions, both before and after centering. As can be seen in *Figures 5b and 6b*, prior to the centering, the network structure is completely dominated by subject-level clustering, with no hint of any task-related structure (i.e., separation of baseline, early, and late learning). However, following the centering procedure, a clear task-related structure is now apparent, particularly between the baseline epoch and the two learning epochs (early and late). Notably, this task-related structure is present in both the cognitive and sensorimotor networks.

Next, we represented these centered covariance matrices as tangent vectors describing differences in functional connectivity (for each task epoch and day) with respect to the mean resting-state scan (see *Varoquaux et al., 2010*). This approach is beneficial, as it has the effect of removing static features of a subject's network structure which do not change across task epochs (and thus are not of interest in the context of learning). We then embedded the ROIs within the cognitive and sensorimotor networks separately into two dimensions using the model of *Wang et al., 2019*, in order to identify components describing patterns of changes in functional connectivity relative to rest. In this embedded space, two ROIs are located close together if they exhibit similar changes in functional connectivity with respect to rest.

Given these components $\mathbf{h}_k$, every tangent vector $\mathbf{T}_i$ is then approximated by $\mathbf{T}_i = \sum_k \mathbf{h}_k^\top \mathbf{h}_k \lambda_{ik}$. The matrices $\mathbf{h}_k^\top \mathbf{h}_k$ can be interpreted as patterns of changes in functional connectivity relative to rest, while the score $\lambda_{ik}$ is the degree to which that pattern is expressed in the $i$th observation. By examining changes in the component scores across group, day, and task epoch, we are able to characterize learning-related changes in cognitive and sensorimotor network structure.

## Changes in cognitive and sensorimotor network structure during learning

*Figure 5* shows the results of the embedding of the cognitive networks. *Figure 5c* shows the grand mean (across all subjects) resting-state covariance matrix to which subject task data were centered, revealing a clear clustering of the FP, DA, and VA networks, consistent with their designation as distinct functional networks. *Figure 5d* shows the joint embedding of all the cognitive network tangent vectors (across subjects, epochs, and days) into two dimensions, resulting in components describing patterns of changes in connectivity relative to rest. The first component appears to be a gradient separating the FP and DA networks, and the second appears to be a general cortical gradient spanning large portions of FP and DA networks. Notably, the VA network loads relatively weakly on both of these components, suggesting that connectivity both within and between this network may not be strongly modulated by task. In order to assess whether these components – and in particular, the latter component, which spans large portions of the cortex – are simply reflective of brainwide changes in the BOLD signal associated with learning, we examined the mean BOLD amplitude across the learning block, as well as within each epoch (baseline, early, and late learning). The data are shown in *Figure 5—figure supplement 1*. In particular, we find no clear trend in the BOLD signal that would account for our components, suggesting that these effects are not merely due to global changes in the BOLD signal.

For each component, we computed the mean component score for each learning group and task epoch. These scores are shown in *Figure 5e*, where the arrows denote the passage of time (from baseline, to early, to late learning), revealing, quite strikingly, that the FF and SF groups show almost identical trajectories on both days, distinct from the SS group. These components are examined separately in *Figure 5f*, where the leftmost panel shows each component in the brain, with each ROI color-coded according to its loading (as in *Figure 5d*).

In order to make the interpretation of the components more clear, it is useful to examine the matrix $\mathbf{h}^\top \mathbf{h}$ for each component $\mathbf{h}$, which can be interpreted as a pattern of change in covariance during each task epoch, relative to rest (*Figure 5c*). *Figure 5g* shows the mean expression of each of these components during each day and task epoch. The first component can be seen to encode a decrease in functional connectivity within the DA network, along with an increase in functional connectivity between the DA and the FP networks. This effect appears to be strongest during the early learning epoch on both days, and to fade during the late learning epoch. In order to examine learning-related changes

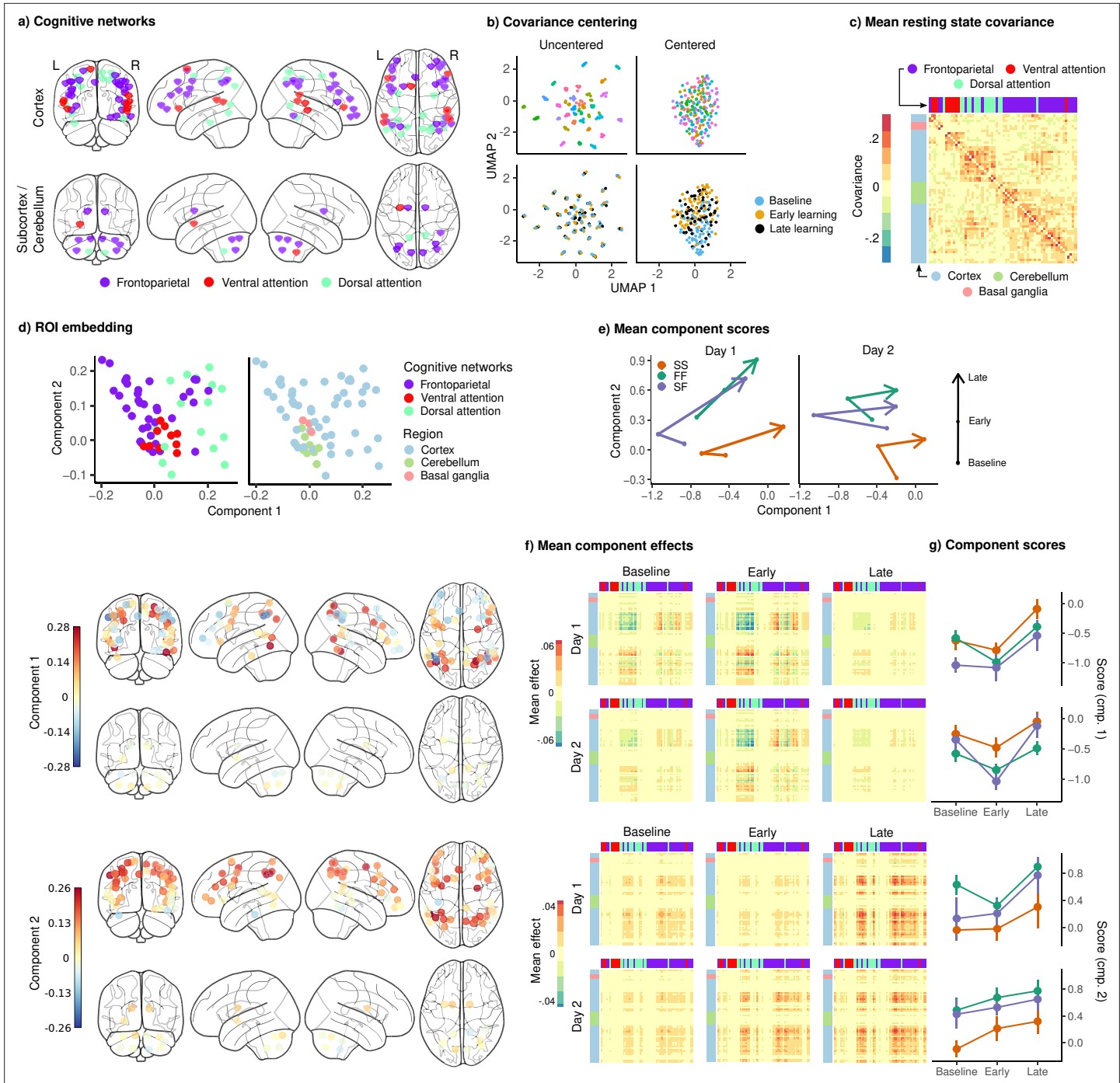

**Figure 5.** Joint embedding of cognitive networks. (**a**) Regions of interest (ROIs) comprising the cognitive networks, colored by the functional network assignment of *Power et al., 2011*. (**b**) Visualization of covariance matrices by uniform manifold approximation (UMAP; *McInnes et al., 2018*). Each dot represents a single covariance matrix. Covariance matrices are shown both before and after centering, and are colored by subject (top) and scan (bottom). Note the strong subject-level clustering in each network before centering, which masks differences in task structure. (**c**) Grand mean resting-state covariance (across all subjects), to which subject task data were centered. Matrix was ordered using single-linkage hierarchical clustering. (**d**) (Top) Components extracted by the embedding. ROIs are colored by functional network assignment (left) and anatomical location (right). (Bottom) Embedding components displayed on the brain. ROIs are colored by their loading on each component. (**e**) Mean component score in each group for each task epoch and day. Arrow denotes the direction of time (baseline to early learning to late learning). (**f**) Components displayed as tangent vectors, reflecting changes in functional connectivity relative to rest. Each matrix represents the overall mean effect in each task epoch and day. All matrices are ordered identically to the grand mean resting-state covariance, for ease of comparison. (**g**) Components scores for each group during each task epoch and day. Error bars denote standard errors.

*Figure 5 continued on next page*

*Figure 5 continued*

The online version of this article includes the following figure supplement(s) for figure 5:

**Figure supplement 1.** Mean BOLD signal and embedding component scores.

in the expression of this component, we fit a mixed-effects ANOVA, with epoch and day as within-subjects factors, to the component scores. Due to the large number of possible comparisons and concerns for power given the small group sizes, we examined only the main effects of group, day, and epoch, and did not perform post hoc testing. *F*-tests were conducted for each main effect, corrected for multiple comparisons (*Benjamini and Hochberg, 1995*), and revealed a significant effect of epoch ($F_{2,29} = 34.65$, p < .001, q < .001). The second component encodes an increase in connectivity within and between the DA and FP networks, and also shows a significant main effect of epoch ($F_{2,29} = 9.45$, p < .001, q < .001), as well as group ($F_{2,29} = 4.38$, p = .022, q = .032). The expression of this component increases from baseline, to early, to late learning, and is expressed more strongly in the FF and SF groups than in the SS group.

The embedding of the sensorimotor networks is shown in *Figure 6*, in identical format to *Figure 5*. The mean resting-state covariance matrix is shown in *Figure 6c*, showing strong connectivity within the SMD network. The embedding of the ROIs in *Figure 6d* reveals a component with positive loadings across almost all of the cortical sensorimotor network ROIs, and a second component separating the dorsal and lateral cortical somatomotor ROIs. Unlike in the cognitive networks, these components do not appear to clearly distinguish the groups, although they do appear to show consistent changes across task epochs (*Figure 6e*).

In *Figure 6f*, we see that the first component encodes the overall change in connectivity with the dorsal somatomotor regions, with baseline and early learning epochs showing a global decrease relative to resting state, which either disappears or reverses during late learning. This trend was present for all groups on both days, and a mixed-effects ANOVA showed a significant main effect of epoch on the expression of this component ($F_{2,29} = 21.71$, p < .001, q < .001).

The second sensorimotor component encodes an increase in connectivity within each of the dorsal and lateral somatomotor networks, and a decrease in connectivity between them. Although this component showed a significant main effect of epoch ($F_{2,29} = 20.19$, p < .001, q < .001), we could not discern any clear pattern to the effect, other than what appears to be a decrease during early learning as compared to baseline, which may suggest a reduced functional segregation between the dorsal and lateral regions during this epoch.

Taken together, these results suggest that, despite similarities in early learning performance between the SF and SS subjects, the SF subjects appear to engage their cognitive systems in a different fashion from the SS subjects over the course of learning. In particular, consistent with the idea that the FF and SF recruit their cognitive systems in a similar fashion, we find that both groups show greater functional connectivity within and between the DA and FP networks (cognitive component 2).

## Discussion

Prior work has shown that some neural activity patterns are more difficult to generate than others, and that this constrains learning *Sadtler et al., 2014*. Here, using human fMRI and by investigating changes in cognitive and sensorimotor network structure during sensorimotor adaptation, we asked whether similar principles might also account for individual differences in learning. Clustering of subject learning data revealed distinct profiles of learning, including a group which adapted rapidly on both days, achieving near-complete reduction of error during late learning, as well as rapid de-adaptation (FF), and a group which adapted and de-adapted slowly on both days (SS). The performance patterns exhibited by the FF group, including rapid learning, unlearning, and relearning, are consistent with the use of an explicit cognitive strategy (*Leow et al., 2017*; *de Brouwer et al., 2018*). Notably, we also observed a third group of subjects (SF), who displayed large early errors and slow adaptation during the first day, but also showed comparable rates of de-adaptation to the FF group, and rapid readaptation on the second day. We interpreted this as evidence that the SF group may have been attempting the use of cognitive re-aiming strategies during initial learning, but that it took them longer to figure out the appropriate strategy. Consistent with this idea, the embedding of subject cognitive networks revealed that network structure in the FF and SF groups were highly similar during learning across

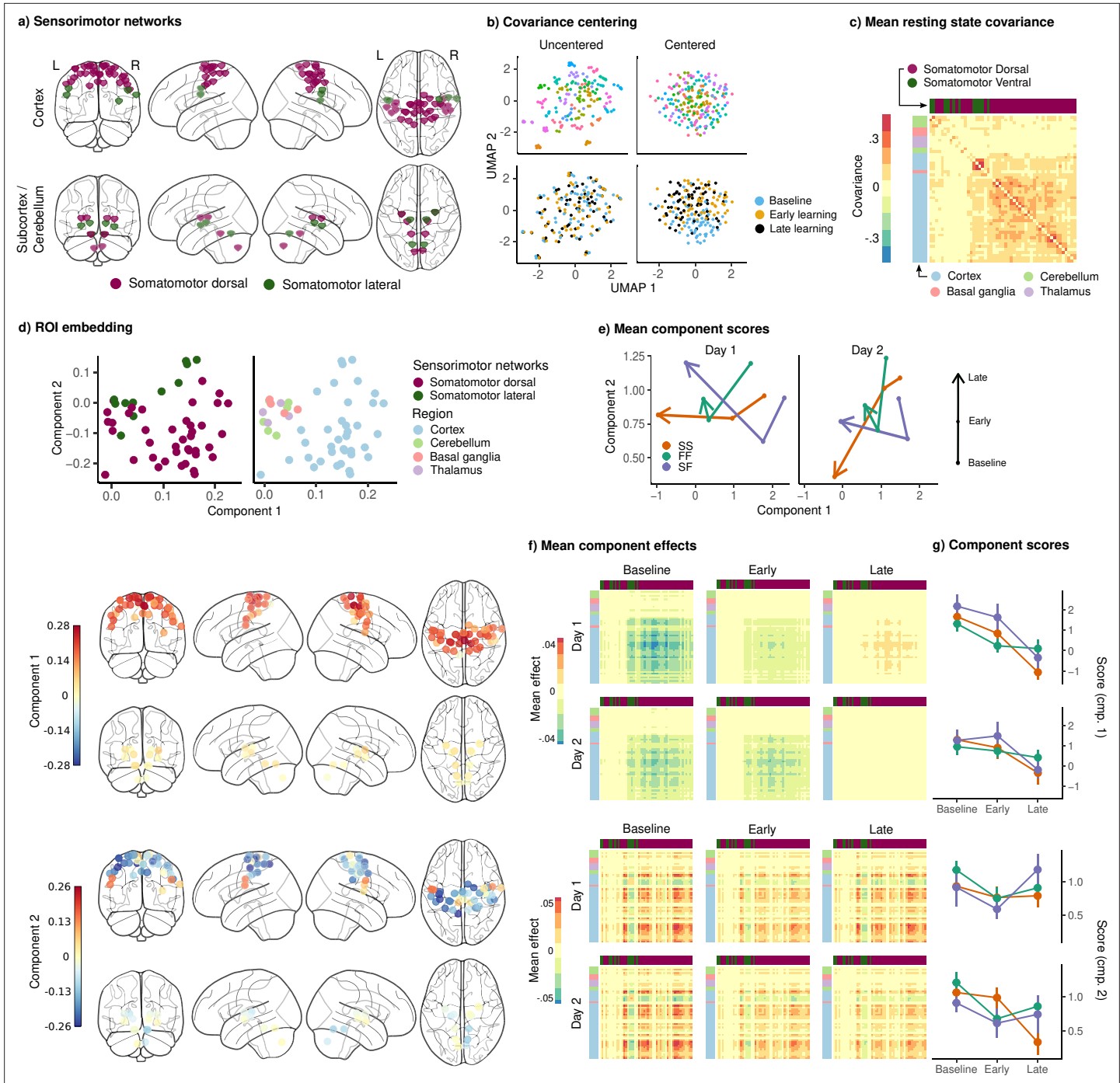

**Figure 6.** Joint embedding of sensorimotor networks. (**a**) Regions of interest (ROIs) comprising the sensorimotor networks, colored by the functional network assignment of *Power et al., 2011*. (**b**) Visualization of covariance matrices by uniform manifold approximation (UMAP; *McInnes et al., 2018*). Each dot represents a single covariance matrix. Covariance matrices are shown both before and after centering, and are colored by subject (top) and scan (bottom). Note the strong subject-level clustering in each network before centering, which masks differences in task structure. (**c**) Grand mean resting-state covariance (across all subjects), to which subject task data were centered. Matrix was ordered using single-linkage hierarchical clustering. (**d**) (Top) Components extracted by the embedding. ROIs are colored by functional network assignment (left) and anatomical location (right). (Bottom) Embedding components displayed on the brain. ROIs are colored by their loading on each component. (**e**) Mean component score in each group for each task epoch and day. Arrow denotes the direction of time (baseline to early learning to late learning). (**f**) Components displayed as tangent vectors, reflecting changes in functional connectivity relative to rest. Each matrix represents the overall mean effect in each task epoch and day. All matrices are ordered identically to the grand mean resting-state covariance, for ease of comparison. (**g**) Components scores for each group during each task epoch and day. Error bars denote standard errors.

both days (*Figure 5e*). In particular, both groups showed high expression of a component encoding an increase in connectivity within and between the FP and DA networks. Our excursion analysis further revealed that these groups' cognitive networks exhibited greater excursion from intrinsic network structure during the learning phase, after onset of the VMR. Taken together, these results suggest that both the FF and SF groups, who display behavioral characteristics consistent with the use of explicit, cognitive strategy use, engage their cognitive brain networks in a manner that is distinct from gradual learners, and to a greater degree.

## Cognitive and sensorimotor network structure during learning

Our cognitive network embedding revealed a pair of components encoding, predominantly, connectivity within and between FP and DA networks. The first of these components appeared to encode a decrease in connectivity within the DA network but an increase in connectivity between the FP and DA networks. This component showed a decrease in expression from early to late learning in all three behavioral groups, suggesting that it may reflect a general feature of learning or task performance (e.g., decreasing errors). The second component, encoding an increase in connectivity both within and between the FP and DA networks, was more strongly expressed in both the FF and SF groups than in the SS group. The DA network in particular is implicated in the top-down orientation of spatial attention (*Simpson et al., 2011*; *Corbetta and Shulman, 2002*), and is known to be modulated by the FP network during goal-directed cognition (*Spreng et al., 2013*). The increase in connectivity within and between these networks in the FF and SF groups may then reflect the deliberate reorienting of attention during task performance, which would be consistent with the finding of *de Brouwer et al., 2018*, that explicit learners alter their gaze behavior to reflect their intended aim direction in response to a VMR. The DA network has also been implicated in motor inhibition (*Hsu et al., 2020*), which in the VMR task is thought to be essential for the initial suppression of a motor response to the actual target prior to implementation of a re-aiming strategy (*Drummond et al., 2015*). The reduced connectivity in the DA network in the SS compared to the FF/SF groups might then reflect weaker engagement of this system. This would also be consistent with the shorter response times in SS, as this initial inhibition and the successful deployment of a re-aiming strategy are believed to require longer preparation (*Fernandez-Ruiz et al., 2011*).

It is notable that subcortical and cerebellar regions loaded relatively weakly on the components extracted in both the cognitive and sensorimotor networks. This, however, should not be taken to suggest that these regions are not functionally engaged during learning, as the cerebellum in particular is known to play a key role in sensorimotor adaptation. Rather, because our components reflect *changes* in functional connectivity with respect to rest, it may suggest that the ways in which these regions are situated in the cognitive and sensorimotor networks are similar across task and rest, or that changes in functional connectivity are relatively larger within cortical networks. In our cognitive network, both components reflected changes in connectivity within and between cortical FP and DA regions, which may suggest that these regions drive the majority of functional reconfiguration during our task.

## Neural excursions from manifold structure

Low-dimensional structure in the activity of local neural populations is generally believed to result from the underlying network organization, with structural connections imposing strong constraints on the activity patterns expressible by the network. Several authors, such as *Sadtler et al., 2014*, have suggested that this structure may, in turn, impose constraints on learning, such that behaviors requiring the generation of on-manifold activity are learned more quickly than behaviors requiring novel, off-manifold activation. Although whole-brain fMRI BOLD activity is likewise known to display low-dimensional structure (e.g., *Gao et al., 2020*), its origins are likely to involve not only the underlying structural connectivity, but also shared function, shared sensory input, and correlations introduced by shared activation during cognitive tasks. In the context of large-scale brain networks, then, excursion from resting manifold structure is likely to reflect, at least partly, a change in the functional organization of these networks.

We observed a large spike in excursion after rotation onset, located predominantly in the cognitive networks, and which was largest on the first day. This spike was most prominent in the FF and SF groups, which show behavioral traits consistent with explicit learning. The excursion does not appear

to be a response to error, as the SS group showed less cognitive excursion than both the FF and SF groups, despite greater errors on both days. Nor does it appear to reflect the pure recall of a strategy itself, as the spike was dampened in the FF group on the second day, while being maintained in the SF group. This may suggest that the excursion reflects cognitive network engagement involved in the learning or refinement of an explicit strategy, and not in the recall or implementation of the strategy itself.

## Methodological considerations

Our characterization of brain network structure during VMR learning comes with a few caveats. First, we consider cognitive and sensorimotor networks separately, while ignoring possible functional interactions between them. In truth, sensorimotor regions receive substantial top-down modulation from higher-order cognitive regions, and these connections are likely essential for successful learning and performance (*Mars et al., 2011*). Our decision to treat these networks separately was motivated largely by interpretability: Whole-brain networks spanning multiple functional domains are more difficult to analyze and summarize, and because our hypotheses concerned the role of higher-order cognitive regions specifically, their separation from sensorimotor regions allowed us to more directly probe the structure of these networks during learning. It is worth noting that our fPCA (*Figure 4b*) of subject excursion revealed higher excursion in sensorimotor as well as cognitive networks in FF and SF learners, which suggests a functional reconfiguration of sensorimotor network structure in explicit learners. This may itself be partly a consequence of a change in cognitive networks, and a resulting change in the functional connections between cognitive and sensorimotor regions.

Second, while our joint embedding and our excursion measure probe the static and dynamic structure of cognitive and sensorimotor networks during learning, respectively, they do so in slightly different ways, and their respective conclusions are difficult to directly compare. Specifically, our excursion measure isolates the off-manifold component of the BOLD signal, and so quantifies the degree to which a brain network exhibits covariance patterns qualitatively different from rest; though it doesn't provide a characterization of the nature of the difference (though this is also true in the neural manifold work of, e.g., *Sadtler et al., 2014*). The embedding, by contrast, is conducted on the centered tangent vectors encoding differences in covariance relative to rest, which include both on-manifold (a change in expression of components already present during rest) and off-manifold components. It is not necessarily the case, then, that the components extracted by our embedding describe the changes in network structure underlying the increase in excursion during learning. Future work will be required to establish this relationship. Moreover, excursion allows for a moment-by-moment characterization of changes in network structure, whose time course may not be fully captured by the windows we have used for our embedding.

Finally, although we interpret several features of subject performance, such as rapid learning, savings, and long reaction times, as reflecting explicit, cognitive learning, our study lacked a direct assay of subjects' explicit strategy. This is often assayed through the incorporation of report trials (*Taylor et al., 2014*), in which subjects are periodically asked to report their aim direction prior to performing a movement. The difference between the target location and reported aim direction then provides a direct measure of a subject's explicit knowledge, which might have allowed stronger claims about the relationship between explicit strategy use and the structure of cognitive brain networks. However, there is considerable evidence (e.g., *Leow et al., 2017*; *de Brouwer et al., 2018*) that these trials can alter subject performance on the task and magnify the use of these strategies. Moreover, their inclusion may have contaminated the neuroimaging data, which would then reflect not only the performance of the main task, but also the secondary reporting task. For these reasons, report trials were not included. Nevertheless, subject reports strongly correlate both with our measures of error and other assays of explicit learning (*Leow et al., 2017*; *de Brouwer et al., 2018*), and measures such as savings and learning rate are linked to explicit learning processes by a large literature, both behavioral and computational (e.g., *McDougle et al., 2015*; *Morehead et al., 2015*).

## Materials and methods

### Subjects

Forty right-handed individuals between the ages of 18 and 35 (M = 22.5, SD = 4.51; 13 males) participated in the study and received financial compensation for their time. Data from eight participants were excluded due to either head motion in the MRI scanner ($N = 4$; motion greater than 2 mm or 2° rotation within a single scan) or their inability to learn the rotation task ($N = 4$), leaving 32 participants in the final analysis. Right-handedness was assessed with the Edinburgh handedness questionnaire (*Oldfield, 1971*). Participants' written, informed consent was obtained before commencement of the experimental protocol. The Queen's University Research Ethics Board approved the study and it was conducted in accordance with the principles outlined in the Canadian Tri-Council Policy Statement on Ethical Conduct for Research Involving Humans and the principles of the Declaration of Helsinki (*World Medical Association, 1964*).

### VMR task

#### Apparatus

Participants performed hand movements directed toward a target by applying a directional force onto an MRI-compatible force sensor (ATI technologies) using their right index finger and thumb. The target and cursor stimuli were rear-projected with an LCD projector (NEC LT265 DLP projector, 1024 × 768 resolution, 60 Hz refresh rate) onto a screen mounted behind the participant. The stimuli on the screen were viewed through a mirror fixed to the MRI coil directly above participants' eyes, thus preventing participants from being able to see the hand. The force sensor and associated cursor movement were sampled at 500 Hz.

#### Procedure

This experiment used a well-established VMR task (*Krakauer, 2009*) to probe sensorimotor adaptation. To start each trial, the cursor (20-pixel radius) appeared in a central start position (25-pixel radius). A white target circle (30-pixel radius) appeared at one of eight locations (0°, 45°, 90°, 135°, 180°, 225°, 270°, 315°) on an invisible ring around the central position (300-pixel radius) and filled in (white) following a 200 ms delay. Once launched, the cursor would travel the 300-pixel distance to the ring over a 750 ms period (with a bell-shaped velocity profile) before becoming stationary at the ring to provide participants with end-point error feedback. If the cursor overlapped with the target to any extent, the target would become green, signifying a target 'hit'. Each trial was separated by 4 s and within this period, participants had 2.6 s from target presentation to complete the trial (including the 200 ms target delay, participants' own reaction time, and the 750 ms cursor movement; any remaining time was allotted to providing the end-point error feedback). At 2.6 s the trial was ended, the screen was blanked, the data saved, and participants would briefly wait for the next trial to begin. Reaction times were not stressed in this experimental procedure. On trials in which the reaction time exceeded 2.6 s, the trial would end, and the participant would wait for the next trial to begin. These discarded trials were rare (0.56% across all trials, all participants) and were excluded from behavioral analyses, but were kept in the neuroimaging analysis due to the continuous nature of the fMRI task and our focus on functional connectivity analyses.

During each testing session, 120 baseline trials (15 sets of eight trials) were completed without a rotation of the cursor. Following these trials, 320 learning trials (40 sets of eight trials) were completed, wherein a 45° clockwise rotation of the cursor was applied. The baseline and learning trials were completed during one continuous fMRI scan. Following this scan, conditions were restored to baseline (i.e., no rotation of cursor) in a separate scan and participants performed 120 washout trials. These washout trials allowed us to probe participants' rate of relearning 24 hr later (and thus, their savings). In addition to these VMR-related task components, we also interspersed three 6 min resting-state fMRI scans prior to, between, and following VMR learning and washout. During these resting-state scans, participants were instructed to rest with their eyes open, while fixating a central cross location presented on the screen. The total testing time was 75 min on each testing day.

## Behavioral data processing

On each trial we measured the angle error between the target and the final cursor position. Trials with reactions times greater than 2 s or less than 100 ms were discarded (the latter value was chosen as a conservative threshold on prepotent or anticipatory responses). Additionally, because subjects occasionally had difficulty utilizing the force sensor, leading to trials in with extreme cursor deviations (e.g., in the opposite direction of the target), we performed outlier removal by first detrending each block by subtracting a smoothed error curve estimated using a cubic spline basis with one knot per 40 trials, and then excluding those trials more than 3 standard deviations from the fitted curve. Excluded trials were then interpolated using the same cubic spline fit. In total, 4.3% of trials were removed and interpolated across all subjects and all task blocks.

## Behavioral summary measures

For each subject, we computed the mean early error during the rotation block on each day, defined as the circular mean error during the first four blocks of eight trials. We defined savings as the difference between early error on days 1 and 2 (*Haith et al., 2015*; *Morehead et al., 2015*). These measures (day 1 early error, day 2 early error, and savings) were clustered using hierarchical complete linkage clustering on the Manhattan distance between the standardized variables. We extracted three clusters, corresponding roughly to subjects which learned the rotation rapidly on both days (FF learners), subjects which learned the rotation slowly on both days (SS), and subjects which learned slowly on day 1 and quickly on day 2 (SF).

We assessed the significance of this cluster solution through a parametric bootstrap procedure inspired by *Kimes et al., 2017*. We drew 5000 bootstrap samples from a null model in which the data were derived from a single multivariate normal distribution with covariance equal to that of the observed data. For each of these bootstrap samples, we performed hierarchical clustering and evaluated the ratio of within cluster to total sums of squares for a number of clusters between 2 and 8 (*Figure 2b*, right). Only solutions with three or four clusters produced within-cluster sums of squares lower than what would be expected under the null model (*Figure 2c*), and as the three-cluster solution was both slightly superior and more parsimonious, this solution formed the basis of our analysis.

## Imaging

### Data collection

Participants were scanned using a 3T Siemens TIM MAGNETOM Trio MRI scanner located at the Centre for Neuroscience Studies, Queen's University (Kingston, Ontario, Canada). For each subject on each day, we collected a T1-weighted ADNI MPRAGE anatomical (TR = 1760 ms, TE = 2.98 ms, field of view = 192 mm × 240 mm × 256 mm, matrix size = 192 × 240 × 256, flip angle = 9°, 1 mm isotropic voxels). fMRI volumes were acquired using a 32-channel head coil and a T2*-weighted single-shot gradient-echo echo-planar imaging acquisition sequence (TR = 2000 ms, slice thickness = 4 mm, in-plane resolution = 3 mm × 3 mm, TE = 30 ms, field of view = 240 mm × 240 mm, matrix size = 80 × 80, flip angle = 90°), and acceleration factor (integrated parallel acquisition technologies) = 2 with generalized auto-calibrating partially parallel acquisitions reconstruction. Each volume comprised 34 contiguous (no gap) oblique slices acquired at a 30° caudal tilt with respect to the plane of the anterior and posterior commissure, providing whole-brain coverage of the cerebrum and cerebellum. For each resting-state scan, 180 imaging volumes were collected. For the baseline and learning epochs, we collected a single, continuous scan of 896 imaging volumes. For the washout scan, we collected one scan of 256 imaging volumes. Each scan included an additional eight imaging volumes at both the beginning and the end.

All imaging and behavioral data are publically available in an OpenNeuro repository (*Standage et al., 2022*, accession number ds004021).

## fMRI preprocessing

Preprocessing was performed using *fMRIPrep* 1.4.0 (*Esteban et al., 2018b*; *Esteban et al., 2018a*; RRID:SCR_016216), which is based on *Nipype* 1.2.0 (*Gorgolewski et al., 2011*; *Gorgolewski et al., 2018*; RRID:SCR_002502).

## Anatomical data preprocessing

T1-weighted (T1w) images were corrected for intensity non-uniformity (INU) with N4BiasFieldCorrection (*Tustison et al., 2010*), distributed with ANTs 2.2.0 (*Avants et al., 2008*, RRID:SCR_004757). The T1w reference was then skull-stripped with a *Nipype* implementation of the antsBrainExtraction.sh workflow (from ANTs), using OASIS30ANTs as target template. Brain tissue segmentation of cerebrospinal fluid (CSF), white matter (WM), and gray matter (GM) was performed on the brain-extracted T1w using fast (FSL 5.0.9, RRID:SCR_002823, *Zhang et al., 2001*). A T1w-reference map was computed after registration of 2 T1w images (after INU correction) using mri_robust_template (FreeSurfer 6.0.1, *Reuter et al., 2010*). Brain surfaces were reconstructed using recon-all (FreeSurfer 6.0.1, RRID:SCR_001847, *Dale et al., 1999*), and the brain mask estimated previously was refined with a custom variation of the method to reconcile ANTs-derived and FreeSurfer-derived segmentations of the cortical GM of Mindboggle (RRID:SCR_002438, *Klein et al., 2017*). Volume-based spatial normalization to two standard spaces (MNI152NLin6Asym, MNI152NLin2009cAsym) was performed through nonlinear registration with antsRegistration (ANTs 2.2.0), using brain-extracted versions of both T1w reference and the T1w template. The following templates were selected for spatial normalization: *FSL's MNI ICBM 152 non-linear 6th Generation Asymmetric Average Brain Stereotaxic Registration Model* (*Evans et al., 2012*, RRID:SCR_002823; TemplateFlow ID: MNI152NLin6Asym), *ICBM 152 Nonlinear Asymmetrical template version 2009c* (*Fonov et al., 2009*, RRID:SCR_008796; TemplateFlow ID: MNI152NLin2009cAsym).

## Functional data preprocessing

For each scan, the following preprocessing was performed. First, a reference volume and its skull-stripped version were generated using a custom methodology of *fMRIPrep*. The BOLD reference was then co-registered to the T1w reference using bbregister (FreeSurfer) which implements boundary-based registration (*Greve and Fischl, 2009*). Co-registration was configured with nine degrees of freedom to account for distortions remaining in the BOLD reference. Head motion parameters with respect to the BOLD reference (transformation matrices, and six corresponding rotation and translation parameters) are estimated before any spatiotemporal filtering using mcflirt (FSL 5.0.9, *Jenkinson et al., 2002*). BOLD runs were slice-time-corrected using 3dTshift from AFNI 20160207 (*Cox and Hyde, 1997*, RRID:SCR_005927). The BOLD time-series, were resampled to surfaces on the following spaces: *fsaverage*. The BOLD time-series (including slice-timing correction when applied) were resampled onto their original, native space by applying a single, composite transform to correct for head motion and susceptibility distortions. These resampled BOLD time-series will be referred to as *preprocessed BOLD in original space*, or just *preprocessed BOLD*. The BOLD time-series were resampled into several standard spaces, correspondingly generating the following *spatially normalized, preprocessed BOLD runs*: MNI152NLin6Asym, MNI152NLin2009cAsym. First, a reference volume and its skull-stripped version were generated using a custom methodology of *fMRIPrep*. Automatic removal of motion artifacts using independent component analysis (*Pruim et al., 2015*) was performed on the *preprocessed BOLD on MNI space* time-series after removal of non-steady-state volumes and spatial smoothing with an isotropic, Gaussian kernel of 6 mm FWHM (full-width half-maximum). Corresponding 'non-aggresively' denoised runs were produced after such smoothing. Additionally, the 'aggressive' noise regressors were collected and placed in the corresponding confounds file. Several confounding time-series were calculated based on the *preprocessed BOLD*: framewise displacement (FD), DVARS, and three region-wise global signals. FD and DVARS are calculated for each functional run, both using their implementations in *Nipype* (following the definitions by *Power et al., 2014*). The three global signals are extracted within the CSF, the WM, and the whole-brain masks. Additionally, a set of physiological regressors were extracted to allow for component-based noise correction (*CompCor*, *Behzadi et al., 2007*). Principal components are estimated after high-pass filtering the *preprocessed BOLD* time-series (using a discrete cosine filter with 128 s cut-off) for the two *CompCor* variants: temporal (tCompCor) and anatomical (aCompCor). tCompCor components are then calculated from the top 5% variable voxels within a mask covering the subcortical regions. This subcortical mask is obtained by heavily eroding the brain mask, which ensures it does not include cortical GM regions. For aCompCor, components are calculated within the intersection of the aforementioned mask and the union of CSF and WM masks calculated in T1w space, after their projection to the native space of each functional run (using the inverse BOLD-to-T1w transformation). Components

are also calculated separately within the WM and CSF masks. For each CompCor decomposition, the $k$ components with the largest singular values are retained, such that the retained components' time-series are sufficient to explain 50% of variance across the nuisance mask (CSF, WM, combined, or temporal). The remaining components are dropped from consideration. The head motion estimates calculated in the correction step were also placed within the corresponding confounds file. The confound time-series derived from head motion estimates and global signals were expanded with the inclusion of temporal derivatives and quadratic terms for each (*Satterthwaite et al., 2013*). Frames that exceeded a threshold of 0.5 mm FD or 1.5 standardized DVARS were annotated as motion outliers. All resamplings can be performed with *a single interpolation step* by composing all the pertinent transformations (i.e., head motion transform matrices, susceptibility distortion correction when available, and co-registrations to anatomical and output spaces). Gridded (volumetric) resamplings were performed using antsApplyTransforms (ANTs), configured with Lanczos interpolation to minimize the smoothing effects of other kernels (*Lanczos, 1964*). Non-gridded (surface) resamplings were performed using mri_vol2surf (FreeSurfer).

Many internal operations of *fMRIPrep* use *Nilearn* 0.5.2 (*Abraham et al., 2014*, RRID:SCR_001362), mostly within the functional processing workflow. For more details of the pipeline, see the section corresponding to workflows in *fMRIPrep*'s documentation.

## Excursion

We defined motor and cognitive networks using the whole-brain parcellation proposed by *Seitzman et al., 2020*, comprising 300 ROIs across the cortex, subcortex, and cerebellum. Our motor network was comprised of those ROIs functionally assigned to the SomatomotorDorsal and Somatomotor-Lateral networks, while our cognitive network was comprised of the FP, DA, and VA networks. The latter was chosen to include a wide array of higher-order association regions in the frontal and parietal cortices. Mean time courses were extracted for each ROI by intersecting an 8 mm sphere with a GM mask, excluding those voxels lying in the intersection of multiple spheres. See *Figures 5a and 6a* for visualizations of each network.

## Network excursion

For each subject, we estimated resting-state manifolds for each of the motor and cognitive networks by applying PCA to the ROI time courses from first resting-state scan on day 1, retaining the number of components required to explain 75% of the variance. Note that we experimented with several procedures for selecting the number of components, including explained variance from 50% to 90%, and the use of *k*-fold cross-validation. As all methods returned similar results, we report the results with a threshold of 75% as a median compromise. The resting-state scan was chosen over the baseline epoch of the task in order to avoid the following possible confound: As the end of the baseline block corresponds precisely with the onset of rotation, it is difficult to disentangle a change in excursion due to (1) introduction of the VMR itself versus (2) the jump expected due to the fact that we are now considering out-of-sample imaging volumes. Because PCA maximizes variance explained *in-sample*, we would expect larger excursions in out-of-sample observations, even if they were generated by the same process. That is, we would expect an increase in excursion during the learning block, even if network structure remained constant.

The components retained from the resting-state scan form a basis for a subspace containing the majority of variability in the observed resting BOLD activation. Assembling these components in the columns of a matrix $\mathbf{V}$, and letting $\mathbf{x}_t$ be the vector of BOLD activation at time $t$, we define the on-manifold to be the projection of $\mathbf{x}_t$ onto the the subspace spanned by $\mathbf{V}$, obtained by the projection matrix

$$\mathbf{P}^{\mathrm{on}} = \mathbf{V}^\top (\mathbf{V}^\top \mathbf{V})^{-1} \mathbf{V}. \tag{1}$$

The off-manifold activation is then the projection onto the orthogonal compliment of this subspace, given by the projection matrix

$$\mathbf{P}^{\mathrm{off}} = \mathbf{I} - \mathbf{P}^{\mathrm{on}}. \tag{2}$$

The on- and off-manifold components of the BOLD signal are then $\mathbf{x}_t^{\mathrm{on}} = \mathbf{P}^{\mathrm{on}}\mathbf{x}_t$ and $\mathbf{x}_t^{\mathrm{off}} = \mathbf{P}^{\mathrm{off}}\mathbf{x}_t$, respectively. We define the off-manifold activation at time $t$ to be $e_t = \|\mathbf{x}_t^{\mathrm{off}}\|$. As this activation is correlated with the overall BOLD activation $\|\mathbf{x}_t\|$, we define the excursion $r_t = e_t/\|\mathbf{x}_t\|$ to be the ratio of the

off-manifold activation to the overall BOLD activation. Intuitively, this measure roughly describes the proportion of the overall BOLD activation which is *off-manifold* (see *Figure 1d*).

After estimating resting-state manifolds for each subject, we computed the excursion within the task scan, comprising both the pre- and post-rotation periods. We then standardized the excursion trajectories using the mean and standard deviation during the pre-rotation trials, to account for baseline differences in excursion.

## Excursion components

For each subject, we derived a set of four excursion trajectories, comprising the excursion at each imaging volume for both days and both networks. We then used mfPCA (*Happ and Greven, 2018*) to identify the dominant patterns of excursion across networks and across days. Just as PCA estimates a basis for a low-dimensional subspace capturing variability in a set of data lying in Euclidean space $\mathbb{R}^n$, so that each observation can be approximated by the sum of a small set of (ideally, interpretable) components, fPCA likewise attempts to approximate a set of *functions* by a smaller set of basis functions. This functional approach is often more suitable in cases where observations represent continuous functions, as the basis expansion used in fPCA allows features like smoothness or periodicity to be encoded into the components. As each subject contributes a set of four observations – excursion trajectories for each day and each network – we sought a *multivariate* decomposition to derive components describing patterns of excursion across days and networks.

We first represented each excursion trajectory as a continuous function using a cubic spline basis with knots at each observation and smoothing penalty selected by generalized cross-validation (*Härdle, 1990*). mfPCA was then performed using the R package MFPCA (*Happ-Kurz, 2020*) using a penalized spline basis. We extracted three components (*Figure 3b*), sufficient to explain the majority of the variance in subject excursion trajectories (*Figure 3c*). These components encoded, respectively, the total excursion in cognitive and sensorimotor networks on day 1; the excursion in the cognitive network on day 1; and the excursion in the cognitive network on day 2. For each component, we compared scores in each group using a Kruskal-Wallis one-way ANOVA. Post hoc comparisons were made using the Conover-Iman procedure (*Conover and Iman, 1979*), corrected for multiple comparisons using the method of *Benjamini and Hochberg, 1995*.

## Network embedding

### Covariance estimation, centering, and embedding

For each subject, we used the estimator proposed by *Ledoit and Wolf, 2004*, to derive covariance matrices for the day 1 rest scan, as well as periods of equivalent length (177 TRs) at the end of baseline, and the beginning and end of rotation (*Figure 1b*). Note that we use only the first resting scan in order to avoid any effects due to learning-related changes in resting-state network structure. For task covariance matrices, we chose periods of equal length to the resting-state scan in order to avoid introducing bias into our centering procedure, as the regularization applied by our covariance estimator is a function of the sample size.

To center the covariance matrices, we took the approach advocated by *Zhao et al., 2018*, which leverages the natural geometry of the space of covariance matrices. We have implemented many of the computations required to replicate the analysis in an publically available R package spdm (*Areshenkoff, 2022a*), which is freely available from a Git repository at https://github.com/areshenk-rpackages/spdm, (copy archived at swh:1:rev:bbb9ea0419092f9cb5bdeaf289d5a691233d8053; *Areshenkoff, 2022a*).

The geometric intuition for the procedure is as follows: We first estimated resting-state and task covariance matrices for each subject, as well as the grand mean covariance matrix across all subjects' resting-state scans. In order to remove static subject difference, we then applied a translation to each subject's covariance matrices so that their resting-state covariance aligned with the grand mean resting-state covariance. For data lying in Euclidean space, this could be accomplished by subtracting, from each covariance matrix, the difference between corresponding subject's resting covariance and the grand mean resting covariance. But because the space of covariance matrices is a Riemannian manifold (and, moreover, because the difference between two covariance matrices is not necessarily a covariance matrix), we must adopt another approach. Intuitively, we flatten out the space in a neighborhood of each subject's resting-state covariance, so that the difference between each task

covariance matrix and resting state is given by a vector (strictly, a symmetric matrix), which can be added and subtracted normally. We then transport these vectors to the grand mean covariance in a way which preserves their direction. We then project these vectors back onto the space of covariance matrices, giving a 'centered' matrix which encodes an identical change in covariance structure, but now with respect to the grand mean. More formally, we project each subject's task matrices onto the tangent space at their resting-state covariance matrix using the matrix logarithm, transport these tangent vectors to the grand mean, and then exponential map back onto the space of covariance matrices. A high-level overview of these techniques is given by *You and Park, 2021*, although we describe the relevant calculations below.

The procedure is as follows. Letting $R_i$ denote the resting-state covariance matrix for subject $i$, we first computed the grand mean resting-state covariance $\bar{R}$ over all subjects using the fixed-point algorithm described by *Congedo et al., 2017*. For each task epoch $j$, we projected the corresponding covariance matrix $S_{ij}$ onto the tangent space at $R_i$ to obtain a tangent vector

$$T_{ij} = R_i^{1/2} \log(R_i^{-1/2} S_{ij} R_i^{-1/2}) R_i^{1/2}, \tag{3}$$

where log denotes the matrix logarithm. This transformation has the effect of linearizing the space around the resting-state scan $R_i$. The tangent vector $T_{ij}$ then encodes the *difference* in covariance between the task epoch $S_{ij}$ and the resting-state scan $R_i$. We then transported each tangent vector to the grand mean $\bar{R}$ using the transport proposed by *Zhao et al., 2018*, obtaining a centered tangent vector $T_{ij}^c$ given by

$$T_{ij}^c = G T_{ij} G^\top, \tag{4}$$

where $G = \bar{R}^{1/2} R_i^{-1/2}$. This centered tangent vector now encodes the same difference in covariance, but now expressed relative to the mean resting-state scan. Finally, we projected each centered tangent vector back onto the space of covariance matrices, to obtain the centered covariance matrix

$$S_{ij}^c = R^{1/2} \exp(R^{-1/2} T_{ij}^c R^{-1/2}) R^{1/2}, \tag{5}$$

where exp denotes the matrix exponential.

In order to visualize the effects of centering, we derived a two-dimensional embedding of the covariance matrices before and after centering using UMAP (*McInnes et al., 2018*) on the matrix of pairwise geodesic distances between covariance matrices (*Figures 5a and 6a*). This embedding reveals that centering was successful in eliminating static differences in covariance and in isolating the effect of task condition.

For the embedding, rather than modelling the centered covariance matrices themselves, we focused our analysis on the centered tangent vectors, which can be interpreted as differences in covariance relative to resting state (*Varoquaux et al., 2010*). We then constructed weighted, signed graphs from these tangent vectors, and derived a joint two-dimensional embedding of the graphs in each network using the model proposed by *Wang et al., 2019*. Intuitively, the embedding seeks a two-dimensional representation of the ROIs in each network, with each individual observation (group, scan, day) receiving a score indicating the degree to which it expresses each component of the embedding. These components are shown in *Figure 5d* and *Figure 6d*, while scores for each observation are given in *Figure 5* and *Figure 6g*.

## Data availability

All behavioral and imaging data (including T1w and functional scans) are hosted in a repository at OpenNeuro (*Standage et al., 2022*, accession number ds004021). Processed behavioral data and ROI time courses are available in a Dryad repository (https://doi.org/10.5061/dryad.h18931znq).

## Code availability

Imaging data were preprocessed using fmriPrep 1.4.0, which is open source and freely available. Operations on covariance matrices, including estimation and centering, were performed using the R package spdm (*Areshenkoff, 2022a*), which is freely available in a repository at https://github.com/areshenk-rpackages/spdm, (copy archived at swh:1:rev:bbb9ea0419092f9cb5bdeaf289d5a691233d8053;

*Areshenkoff, 2022a*). Tutorial code for implementing the centering procedure, excursion analysis, and embedding are hosted in a GitHub repository at https://github.com/areshenk-opendata/2022-vmr-neuralexcursions, (copy archived at swh:1:rev:96c15068e902ecd58646e1d7ba05899a5f8feaa7; *Areshenkoff, 2022b*).

## Additional information

### Funding

| Funder | Grant reference number | Author |
|---|---|---|
| Canadian Institutes of Health Research | MOP126158 | Jason P Gallivan |

The funders had no role in study design, data collection and interpretation, or the decision to submit the work for publication.

### Author contributions

Corson Areshenkoff, Conceptualization, Data curation, Formal analysis, Methodology, Software, Visualization, Writing – original draft, Writing – review and editing; Daniel J Gale, Conceptualization, Data curation, Methodology, Software; Dominic Standage, Conceptualization, Investigation, Methodology; Joseph Y Nashed, Data curation, Investigation, Software; J Randall Flanagan, Conceptualization, Project administration, Resources, Supervision, Writing – review and editing; Jason P Gallivan, Conceptualization, Funding acquisition, Investigation, Methodology, Project administration, Resources, Supervision, Writing – original draft, Writing – review and editing

### Author ORCIDs

Corson Areshenkoff (iD) http://orcid.org/0000-0001-9072-9185

### Ethics

Human subjects: Participants' written, informed consent was obtained before commencement of the experimental protocol. The Queen's University Research Ethics Board approved the study and it was conducted in accordance with the principles outlined in the Canadian Tri-Council Policy Statement on Ethical Conduct for Research Involving Humans and the principles of the Declaration of Helsinki (1964).

### Decision letter and Author response

Decision letter https://doi.org/10.7554/eLife.74591.sa1
Author response https://doi.org/10.7554/eLife.74591.sa2

## Additional files

### Supplementary files

• Transparent reporting form

### Data availability

All behavioral and imaging data (including T1w and functional scans) are hosted in a repository at OpenNeuro (Standage et al., 2022, accession number ds004021). Processed behavioral data and ROI timecourses are available in a Dryad repository (https://doi.org/10.5061/dryad.h18931znq). Imaging data were preprocessed using fmriPrep 1.4.0, which is open source and freely available. Operations on covariance matrices, including estimation and centering, were performed using the R package spdm (Areshenkoff, 2022b), which is freely available in a repository at https://github.com/areshenk-rpackages/spdm, (copy archived at swh:1:rev:bbb9ea0419092f9cb5bdeaf289d5a691233d8053). Tutorial code for implementing the centering procedure, excursion analysis, and embedding are hosted in a GitHub repository at https://github.com/areshenk-opendata/2022-vmr-neuralexcursions, (copy archived at swh:1:rev:96c15068e902ecd58646e1d7ba05899a5f8feaa7, Areshenkoff, 2022a).

The following datasets were generated:

| Author(s) | Year | Dataset title | Dataset URL | Database and Identifier |
|---|---|---|---|---|
| Areshenkoff C | 2022 | Article: Neural excursions from low-dimensional manifold structure explain patterns of learning and relearning during human sensorimotor adaptation | https://doi.org/10.5061/dryad.h18931znq | Dryad Digital Repository, 10.5061/dryad.h18931znq |
| Standage D, Nashed JY, Flanagan JR, Gallivan JP | 2022 | Visuomotor Rotation Adaptation Experiment | https://doi.org/10.18112/openneuro.ds004021.v1.0.0 | OpenNeuro, 10.18112/openneuro.ds004021.v1.0.0 |

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
