## [Editor Report]

This manuscript describes a fascinating experiment looking at gross network dynamics across cognitive and motor circuits and across different stages of learning, during an adaptive visuomotor learning experiment in the MRI environment. The finding of reliable "excursions" from low-dimensional network states that are associated with learning, primarily in cognitive networks, and that this excursion metric is a reliable indicator of differences in learning has strong implications for our understanding of the way macroscopic brain networks learn new skills.

---

## [Decision Letter]

**Decision letter after peer review:**

Thank you for submitting your article "Neural Excursions from Low-Dimensional Manifold Structure Explain Intersubject Variation in Human Motor Learning" for consideration by *eLife*. Your article has been reviewed by 3 peer reviewers, including Timothy Verstynen as the Reviewing Editor and Reviewer #1, and the evaluation has been overseen by Chris Baker as the Senior Editor.

Essential revisions:

This manuscript describes a fascinating experiment looking at gross network dynamics across cognitive and motor circuits and across different stages of learning, during an adaptive visuomotor learning experiment in the MRI environment. The authors found that there were reliable "excursions" from low-dimensional network states that are associated with learning, primarily in cognitive networks, and that this excursion metric is a reliable indicator of differences in learning.

All three reviewers thought this was a very well designed study, with sophisticated analyses designed to ask deep questions about the nature of brain network interactions during motor skill learning. This is reflected in the reviews (shown below). There are a few points of clarification, however, that need to be resolved before publication.

Here I provide a singular consolidated review, along 3 major themes, for guidance with your revisions. You can feel free to organize your replies around the consolidated review.

General concerns

1. Manuscript logic

Reviewer 1 felt that the manuscript describes essentially 2 phases of inquiry: (1) do excursions of network states, during the task, from resting-state baselines, associate with learning? and (2) if so what are the characteristics of these network changes in motor and cognitive networks? Yet the logic of first identifying an effect and then characterizing it is reversed in the manuscript. The authors start by carefully explaining how cognitive and motor subnetworks vary across learning, with cognitive networks having more consistent shifts across groups, and ‘then’ highlight how deviations from resting states correlates with learning and distinguishes learning groups. This seems to put the cart before the horse.

Reviewer 2 also wondered about the restriction to motor and cognitive subnetworks, wondering at times how other networks that might be involved in the task (e.g., visual regions for the perception of the stimuli; or endogenous attentional regions that might have otherwise impacted the performance, and hence would need to be inhibited in order to promote effective performance) may have been associated with the task. Have the authors analysed data outside of the selected ROIs? And do these regions conform to expectation?

2. Groups

Reviewer 1 thought that the stratification of the three learning groups, based on behavioral performance, feels a little post hoc, particularly given the sample size of the study. It is particularly odd given how the excursion metrics are calculated on a per-subject basis, which would be perfect for a true individual differences analysis. Was a subject-wise association analysis performed and failed? If the goal is to look at individual differences in learning driven by excursions from a resting null state, it seems like a subject-by-subject comparison is more intuitive. Of course, if the sample size is too small for such analysis and stratification based on learning style increases sensitivity, then that is fine (it happens with pesky imaging studies). The authors should, at a minimum, address why this was not done.

This was mirrored in Reviewer 3's concerns as well. The reviewer points out that this manuscript sets out to address how brain dynamics underlie individual differences in learning, but these remain addressed largely as two separate questions: (1) what are the differences amongst participants (with compelling finding of distinct subgroups)? (2) what is the underlying brain dynamics. While the latter question is analysed by splitting individuals into subgroups, the results do not present compelling evidence of group differences. Most of the metrics show no differences amongst the groups or very subtle differences (e.g. differences in cognitive network embedding in Figure 3g), which are commonly only descriptive and not backed up with the necessary statistics. This may be because of a relatively small sample size (N=32) which is likely insufficient to address the question of intersubject variation with sufficient power. It is therefore difficult to say if the presented findings are evidence of absence of group differences or absence of evidence for intersubject variation. In light of this, the title of the manuscript seems misleading, as the manuscript does not present convincing evidence of how 'neural excursions from low-dimensional manifold structure explain intersubject variation in human motor learning'. Therefore, the link between intersubject differences and network dynamics should be made stronger to reflect the abstract, title and interpretation, or the two questions (on behavioural differences and network dynamics) should be addressed separately.

3. Analysis

Reviewer 1 felt that the manifold analysis is incredibly interesting and appears to yield fascinating learning dynamics (particularly the mfPCA results in Figure 6). However, a careful read of the methods leads to many questions. For example, it is not intuitively clear how the authors get to the form of the tangent vector in Eq. 1. Also, it is not clear if the weight graphs (Figures 3 & 4) are derived from the tangent vectors (T) or the projection into the covariance matrices (S)? What is "x_t" in the excursion calculations? Is P^{on} just a norm on the PCA components? Why use a mixture of linear and non-linear methods to find low dimensional subspaces? Given the novelty of the approach, understanding these steps is crucial for the reader to interpret the results.

This was echoed in Reviewer 2's concern as to why the authors chose to use UMAP (a non-linear dimensionality reduction technique) to demonstrate the utility of their mean-centering approach (which the reviewer found to be well-justified), but then later used a functional variant of PCA on the actual data themselves.

Along the same lines, Reviewer 1 felt that the resting state is used as the baseline by which deviances are estimated. Wouldn't the Baseline period during the task be a more appropriate baseline for looking at learning related changes? It seems like using rest, without the same sensory or motor engagement, would be the wrong way to isolate excursions from the manifold from learning alone.

This was echoed in Reviewer 3's comments as well. The network embedding analyses are performed by defining regions of interest (across cognitive and sensorimotor networks), followed by calculating the covariance matrix across defined regions in each learning stage, and contrasting the obtained matrices to the group average resting state. The authors justify well why centering to the group average resting state is necessary to remove subject-specific differences and focus on the learning-related changes. Yet, their subsequent assessment across the baseline/early/late stages allows for interpretations of only how the relative differences in the strength of (task)-components differs across learning compared to the resting state. Given the nature of the visuomotor rotation task, wouldn't the initial 'baseline' stage provide a more natural starting point to assess changes that occur with learning? This would also enable a more direct assessment of whether there is a reconfiguration of network dynamics after the onset of rotation, and how it unfolds during subsequent learning.

Reviewer 2 also wondered whether the increase in the diffuse cognitive component, in which covariance was relatively wide-spread across the cortical regions, may have been associated with a decrease in the BOLD signal as a function of improved performance? To this end, it could be useful for the authors to plot the raw BOLD (or β weights following a GLM) collapsed across the loadings of each of the eigenvectors. This would give the reader a sense of how the covariance related to recruitment, as covariance will increase regardless of whether pairs of regions both went up or both went down.

Reviewer 3 had similar concerns with the interpretation of the manifold analyses. The presented analyses on network dynamics are rigorous and justified, yet interpreting some of the presented results is not straightforward. For instance, in Figure 3e, it seems that the mean component effect for embedding of cognitive networks shows an effect where the network landscape initially moves away from 'baseline' to 'early' stage, and then reverses for 'late' stage such that late stage is more similar to baseline than to early stage on component 1. In the manuscript, only a comparison across the groups is interpreted (FF and SF groups showing more similar trajectories), but how does one interpret changes in the trajectories themselves? Could it be that the changes in relative connectivity within this component (the dorsoattentional network and between dorsoattentional and frontoparietal networks) reflects increases in error rate, or differences in reaction time? There is potential to link these network metrics to subject performance which would help in interpreting the findings as well as relate them more closely to variation in performance.

*Reviewer #1 (Recommendations for the authors):*

This manuscript describes a fascinating experiment looking at gross network dynamics across cognitive and motor circuits and across different stages of learning, during an adaptive visuomotor learning experiment in the MRI environment. The authors found that there were reliable "excursions" of network states associated with learning, primarily in cognitive networks, and that this excursion metric is a reliable indicator of differences in learning.

Overall, this is a very well designed study, with sophisticated analyses designed to ask deep questions about the nature of brain network interactions during motor skill learning. There are many strengths here. My critique focuses only on the aspects of the study and manuscript that require clarity.

1. Manuscript logic

The manuscript describes essentially 2 phases of inquiry: (1) do excursions of network states, during the task, from resting-state baselines, associate with learning? and (2) if so what are the characteristics of these network changes in motor and cognitive networks? Yet the logic of first identifying an effect and then characterizing it is reversed in the manuscript. The authors start by carefully explaining how cognitive and motor subnetworks vary across learning, with cognitive networks having more consistent shifts across groups, and *then* highlight how deviations from resting states correlates with learning and distinguishes learning groups. This seems to put the cart before the horse.

2. Groups

The stratification of the three learning groups, based on behavioral performance, feels a little post hoc, particularly given the sample size of the study. It is particularly odd given how the excursion metrics are calculated on a per-subject basis, which would be perfect for a true individual differences analysis. Was a subjectwise association analysis performed and failed? If the goal is to look at individual differences in learning driven by excursions from a resting null state, it seems like a subject-by-subject comparison is more intuitive. Of course, if the sample size is too small for such analysis and stratification based on learning style increases sensitivity, then that is fine (it happens with pesky imaging studies). The authors should, at a minimum, address why this was not done.

3. Analysis

The manifold analysis is incredibly interesting and appears to yield fascinating learning dynamics (particularly the mfPCA results in Figure 6). However, a careful read of the methods leads to many questions. For example, it is not intuitively clear how the authors get to the form of the tangent vector in Eq. 1. Also, it is not clear if the weight graphs (Figures 3 & 4) are derived from the tangent vectors (T) or the projection into the covariance matrices (S)? What is "x_t" in the excursion calculations? Is P^{on} just a norm on the PCA components? Why use a mixture of linear and non-linear methods to find low dimensional subspaces? Given the novelty of the approach, understanding these steps is crucial for the reader to interpret the results.

Along the same lines, resting state is used as the baseline by which deviances are estimated. Wouldn't the Baseline period during the task be a more appropriate baseline for looking at learning related changes? It seems like using rest, without the same sensory or motor engagement, would be the wrong way to isolate excursions from the manifold from learning alone.

*Reviewer #2 (Recommendations for the authors):*

The authors use a set of dimensionality reduction techniques to analyse fMRI data collected while participants performed a sensorimotor adaptation task. They find evidence to suggest that the capacity for participants to learn the sensorimotor contingencies was associated with deviation from low-dimensional manifolds estimated from the data.

The manuscript was clearly presented, and the methodological choices were well-justified. There was a thorough review of the existing literature, focussing on how low-dimensional signatures estimated from neuroimaging data might relate to the capacity to learn sensorimotor contingencies in a research setting. The aims of the project were spelled out clearly, and the methods aligned well with the proposed goals of their study.

One aspect of the study that I enjoyed was the appreciation of the individual differences in learning rates. -While the group mean appeared to show a gradual increase in performance across the cohort, clustering of the trajectories revealed separable patterns of performance improvement which, if analysed en masse, may have led to erroneous conclusions regarding the neurobiological substrates of these effects. One side-effect of this approach is that it did render the Results quite complex, however the authors did a nice job of regularly referring back to the main point of the manuscript when relaying each set of results.

I was a little confused as to why the authors chose to use UMAP (a non-linear dimensionality reduction technique) to demonstrate the utility of their mean-centering approach (which I found to be well-justified), but then later used a functional variant of PCA on the actual data themselves.

I expect that this manuscript will have a positive effect on the field, as it links exciting dimensionality techniques to data recorded across the performance of an interesting cognitive task.

I did find myself wondering at times how other networks that might be involved in the task (e.g., visual regions for the perception of the stimuli; or endogenous attentional regions that might have otherwise impacted the performance, and hence would need to be inhibited in order to promote effective performance) may have been associated with the task. Have the authors analysed data outside of the selected ROIs? And do these regions conform to expectation?

I found myself wondering whether the increase in the diffuse cognitive component, in which covariance was relatively wide-spread across the cortical regions, may have been associated with a decrease in the BOLD signal as a function of improved performance? To this end, it could be useful for the authors to plot the raw BOLD (or β weights following a GLM) collapsed across the loadings of each of the eigenvectors. This would give the reader a sense of how the covariance related to recruitment, as covariance will increase regardless of whether pairs of regions both went up or both went down.

*Reviewer #3 (Recommendations for the authors):*

The present investigation aimed at investigating changes in brain dynamics that underlie motor learning, and how this relates to intersubject differences in learning.

The manuscript presents a learning fMRI study, employing the visuomotor rotation paradigm, which participants performed on two consecutive days. To assess differences in learning ability amongst participants, participants' performance is clustered based on behavioural measures of error and savings, which results in three distinct subgroups of learners. Then, the brain dynamics is assessed by estimating the covariance networks from cognitive and sensorimotor regions across different learning epochs. Several metrics are derived to explain how network dynamics changes with learning, finding especially differences within the 'cognitive' network. These metrics are also separately inspected across the three distinct subgroups with the aim to relate network dynamics to intersubject variation in learning ability.

Strengths:

The question on how brain dynamics change during learning is an important one, of potential interest to a broad audience. The authors utilize advanced and rigorous techniques to address this question. This has in the domain of motor learning, specifically on visuomotor rotation not been done and is therefore an important advancement of the field. Also dividing brain regions into 'motor' and 'cognitive' networks is relevant as several motor learning studies have demonstrated recently that 'motor' learning might be underpinned by substantial changes in regions traditionally regarded as 'cognitive'. Finally, the clustering of participants into distinct groups based on their learning ability is justified and a valid approach, especially for small sample sizes, to aim at investigating differences in learning ability across individuals. Despite many strengths of the manuscript, there are some outstanding concerns, especially pertaining to the interpretation of the link between brain dynamics and individual differences in learning.

Weaknesses:

1) First, this manuscript sets out to address how brain dynamics underlie individual differences in learning, but these remain addressed largely as two separate questions: 1) what are the differences amongst participants (with compelling finding of distinct subgroups), 2) what is the underlying brain dynamics. While the latter question is analysed by splitting individuals into subgroups, the results do not present compelling evidence of group differences. Most of the metrics show no differences amongst the groups or very subtle differences (e.g. differences in cognitive network embedding in Figure 3g), which are commonly only descriptive and not backed up with the necessary statistics. This may be because of a relatively small sample size (N=32) which is likely insufficient to address the question of intersubject variation with sufficient power. It is therefore difficult to say if the presented findings are evidence of absence of group differences or absence of evidence for intersubject variation. In light of this, the title of the manuscript seems misleading, as the manuscript does not present convincing evidence of how 'neural excursions from low-dimensional manifold structure explain intersubject variation in human motor learning'. Therefore, the link between intersubject differences and network dynamics should be made stronger to reflect the abstract, title and interpretation, or the two questions (on behavioural differences and network dynamics) should be addressed separately.

2) The presented analyses on network dynamics are rigorous and justified, yet interpreting some of the presented results is not straightforward. For instance, in Figure 3e, it seems that the mean component effect for embedding of cognitive networks shows an effect where the network landscape initially moves away from 'baseline' to 'early' stage, and then reverses for 'late' stage such that late stage is more similar to baseline than to early stage on component 1. In the manuscript, only a comparison across the groups is interpreted (FF and SF groups showing more similar trajectories), but how does one interpret changes in the trajectories themselves? Could it be that the changes in relative connectivity within this component (the dorsoattentional network and between dorsoattentional and frontoparietal networks) reflects increases in error rate, or differences in reaction time? There is potential to link these network metrics to subject performance which would help in interpreting the findings as well as relate them more closely to variation in performance.

3) The network embedding analyses are performed by defining regions of interest (across cognitive and sensorimotor networks), followed by calculating the covariance matrix across defined regions in each learning stage, and contrasting the obtained matrices to the group average resting state. The authors justify well why centering to the group average resting state is necessary to remove subject-specific differences and focus on the learning-related changes. Yet, their subsequent assessment across the baseline/early/late stages allows for interpretations of only how the relative differences in the strength of (task)-components differs across learning compared to the resting state. Given the nature of the visuomotor rotation task, wouldn't the initial 'baseline' stage provide a more natural starting point to assess changes that occur with learning? This would also enable a more direct assessment of whether there is a reconfiguration of network dynamics after the onset of rotation, and how it unfolds during subsequent learning.

It would be useful if the performed analyses / metrics were better justified or interpreted. For instance, in the functional PCA analysis (Figure 6) what is the interpretation of having a sinusoidal component? Is this in any way linked to the onset of rotation? If not, what would be the rationale of having such a component in the brain dynamics? What could this reflect?

---

## [Author Response]

Essential revisions:General concerns1. Manuscript logicReviewer 1 felt that the manuscript describes essentially 2 phases of inquiry: (1) do excursions of network states, during the task, from resting-state baselines, associate with learning? and (2) if so what are the characteristics of these network changes in motor and cognitive networks? Yet the logic of first identifying an effect and then characterizing it is reversed in the manuscript. The authors start by carefully explaining how cognitive and motor subnetworks vary across learning, with cognitive networks having more consistent shifts across groups, and ‘then’ highlight how deviations from resting states correlates with learning and distinguishes learning groups. This seems to put the cart before the horse.

After considering these points, and some of the others raised by the reviewers, we agree with reviewer 1 that the logical flow of the manuscript is improved with the excursion analysis presented up front, followed by the joint embedding analyses. We have thus reordered these sections in the Results, and framed the embedding analysis more clearly as an effort to characterize the changes in network structure underlying the excursion effects observed in the first section. The introductions to these respective sections have likewise been rewritten to make this structure more clear. We kindly ask the reviewers to see these modifications in the actual manuscript.

Reviewer 2 also wondered about the restriction to motor and cognitive subnetworks, wondering at times how other networks that might be involved in the task (e.g., visual regions for the perception of the stimuli; or endogenous attentional regions that might have otherwise impacted the performance, and hence would need to be inhibited in order to promote effective performance) may have been associated with the task. Have the authors analysed data outside of the selected ROIs? And do these regions conform to expectation?

Reviewer 2 is right that similar questions could be asked of many networks, such as early visual regions essential for the perception of the stimulus and/or sensory feedback processing of errors, or e.g. middle temporal regions that may play a role in the recall of the task or strategy during relearning on the second day. Our decision to focus on sensorimotor and cognitive networks was motivated by two considerations: First, the division between implicit learning systems (supported by cortical and cerebellar sensorimotor systems), and explicit learning systems (supported by higher-order association regions) is clear in the VMR literature, and is essential for characterizing learning in this task. Our networks were thus targeted towards these systems in particular. Second, it was an attempt to simplify the analysis as much as possible (both statistically and conceptually), given the complexity of the data in general, and the limited sample size available in our study. In particular, we wished to avoid having to conduct separate analyses for a large number of networks, and then struggle to summarize the results in a way that was both acceptably powered and interpretable, and that, moreover, would allow us to link our findings to the existing literature on explicit vs. implicit learning processes. Given that the two networks under consideration can be linked to learning systems that are well understood in the context of the task, we felt that this was the appropriate level of complexity that could reasonably address our research question.

Nevertheless, we appreciate that the readers, in addition to this reviewer, might be interested in seeing the effects occurring outside of our selected subnetworks. To this end, as noted at the outset, we have created a new supplemental figure (Figure 3 – supplemental figure 1), wherein we show the excursion profiles on a network-by-network basis for the majority of the task-positive networks in the Seitzmann atlas. This figure includes excursion curves for each network individually, as well as mean excursion during early and late learning. Note that networks are ordered by early excursion in order to most clearly illustrate trends in excursion across networks. This analysis reveals large spikes in excursion after rotation onset in fronto-parietal, ventral-attention, and cingulo-opercular networks, as well as smaller and more sustained increases in excursion in sensorimotor and visual networks. In order to summarize these differences, we conducted a principal component analysis on early and late excursion in each network. We present the scores on each component for each subject, as well as group means, revealing that FF and SF subjects again display similar patterns of excursion, consistent with our analysis in the main manuscript.

To the reviewer's interest in visual regions specifically, note that we observe a sustained increase in excursion in the visual network (comprising V1 and surrounding regions) after rotation onset, most prominently on day 1. This excursion may reflect the perception of the target stimulus itself, or the visual error feedback received by participants, or the top-down modulation of these regions during learning. Any interpretation would, of course, be speculative, but it is nevertheless interesting. Another potential point of interest is the difference in excursion between the dorsal and ventral attention networks, which have frequently been implicated in the top-down and bottom-up orienting of attention, respectively. In our main analysis, we observed increased connectivity between dorsal attention and fronto-parietal networks, consistent with the top-down orienting of spatial attention during learning. Here, examining the two attention networks (dorsal vs. ventral) in isolation (i.e. not considering their interactions with other cognitive regions), we observe the greatest excursion in the ventral attention network. Although we are reluctant to speculate too deeply about these differences, it may be that this network -- which is implicated in the involuntary orientating of spatial attention -- becomes strongly engaged at the onset of the rotation, before being later suppressed during learning.

2. GroupsReviewer 1 thought that the stratification of the three learning groups, based on behavioral performance, feels a little post hoc, particularly given the sample size of the study. It is particularly odd given how the excursion metrics are calculated on a per-subject basis, which would be perfect for a true individual differences analysis. Was a subject-wise association analysis performed and failed? If the goal is to look at individual differences in learning driven by excursions from a resting null state, it seems like a subject-by-subject comparison is more intuitive. Of course, if the sample size is too small for such analysis and stratification based on learning style increases sensitivity, then that is fine (it happens with pesky imaging studies). The authors should, at a minimum, address why this was not done.

The clustering of subject behavior was indeed post-hoc to the observation of the behavior itself, though not to the fMRI analysis. One of the challenges to conducting a more "pure" individual differences analysis is that it is not obvious, in this case, how an appropriate summary measure should be chosen. Subject behavior in the VMR task is often summarized using scalar measures like savings (the difference in early error between learning and relearning), but as we observe in our data, these measures do not do a good job of fully characterizing subject performance. For example, savings itself does not distinguish between subjects who learn rapidly on both days, and those who learn slowly on both days. Nor does early error, another common measure, distinguish between subjects who learn and unlearn slowly, and those who learn slowly but show behavioral characteristics (e.g., higher RTs) associated with explicit learners during late learning.

Characterizing subject performance in this task thus seems to be an inherently multivariate problem. It would be possible to assess subjects on a variety of measures, and correlate our neural measures with each of them separately. Or, alternatively, to conduct a more elaborate multivariate analysis (e.g. association analysis, or perhaps a canonical correlation analysis) in order to identify relationships between behavioral and neural features. But in that case we would indeed be stretching our sample size to the limit (as the reviewer points out), and these analyses would add complexity to an already complex analysis.

We view our clustering analysis as a simplified approach to this problem. We begin with a more complex, multivariate summary of subject behavior, and condense it down to broad categories. These clusters have the benefit of capturing the dominant patterns of variability in the data, of being clearly interpretable, and of reducing the complexity of the problem to a level that can be reasonably tackled at the sample sizes typical of an imaging study. Although we touched on some of these issues in the original version of the manuscript, we have revised the description of the behavioral analysis to make the reasons for these decisions more explicit. In particular, we now state:

“Since a full characterization of subject performance would require summarizing rates of learning, unlearning, and the total adaptation on both days (along with other features of performance, e.g. reaction times, which have been linked to distinct cognitive or motor processes), here we sought an interpretable, low dimensional summary amenable to simple statistical analysis. This approach was motivated by our observation that standard summary measures generally failed to capture overall performance patterns across both days (see also Standage et al., 2020). For example, 'savings', the difference in early learning performance across days, failed to distinguish between subjects who learned the rotation rapidly on both days versus those who learned and relearned slowly. Similarly, early learning did not distinguish between subjects who adapted and deadapted slowly on the first day versus those who learned slowly but showed performance features similar to fast learners by the end of the first day.”

This was mirrored in Reviewer 3's concerns as well. The reviewer points out that this manuscript sets out to address how brain dynamics underlie individual differences in learning, but these remain addressed largely as two separate questions: (1) what are the differences amongst participants (with compelling finding of distinct subgroups)? (2) what is the underlying brain dynamics. While the latter question is analysed by splitting individuals into subgroups, the results do not present compelling evidence of group differences. Most of the metrics show no differences amongst the groups or very subtle differences (e.g. differences in cognitive network embedding in Figure 3g), which are commonly only descriptive and not backed up with the necessary statistics. This may be because of a relatively small sample size (N=32) which is likely insufficient to address the question of intersubject variation with sufficient power. It is therefore difficult to say if the presented findings are evidence of absence of group differences or absence of evidence for intersubject variation. In light of this, the title of the manuscript seems misleading, as the manuscript does not present convincing evidence of how 'neural excursions from low-dimensional manifold structure explain intersubject variation in human motor learning'. Therefore, the link between intersubject differences and network dynamics should be made stronger to reflect the abstract, title and interpretation, or the two questions (on behavioural differences and network dynamics) should be addressed separately.

We thank the reviewers for pointing out this disconnect in the language used throughout our paper (re: intersubject differences) with the actual results presented (i.e., subject clusters). First, we believe that our clustering does a good job of capturing the major patterns of individual differences in subject performance, but we recognize that our analysis is not a "pure" individual differences study, and that some readers may expect a subject-level analysis given the language in our title/abstract. To more faithfully represent the analysis reported in our manuscript, we've changed the language in our title, abstract, and introduction to more accurately reflect that our analysis focuses on patterns of learning and relearning, rather than subject-level performance. For example, we have now updated the title of the paper to read: "Neural Excursions from Low-Dimensional Manifold Structure Explain patterns of learning and relearning during human sensorimotor adaptation". As for the other related changes, we would kindly ask the reviewer's to see these changes in the actual manuscript (flagged in red text).

3. AnalysisReviewer 1 felt that the manifold analysis is incredibly interesting and appears to yield fascinating learning dynamics (particularly the mfPCA results in Figure 6). However, a careful read of the methods leads to many questions. For example, it is not intuitively clear how the authors get to the form of the tangent vector in Eq. 1. Also, it is not clear if the weight graphs (Figures 3 & 4) are derived from the tangent vectors (T) or the projection into the covariance matrices (S)? What is "x_t" in the excursion calculations? Is P^{on} just a norm on the PCA components? Why use a mixture of linear and non-linear methods to find low dimensional subspaces? Given the novelty of the approach, understanding these steps is crucial for the reader to interpret the results.

We recognize that the embedding analysis, and in particular the description of our centering approach is complex, and we have attempted to revise our description of these methods to make the analysis more clear. In particular, we emphasize more clearly that the embedding was conducted using the tangent vectors themselves, and we now describe our notation more explicitly. We would kindly ask the reviewers to see this section of the Methods (wherein the changes are shown in red text).

This was echoed in Reviewer 2's concern as to why the authors chose to use UMAP (a non-linear dimensionality reduction technique) to demonstrate the utility of their mean-centering approach (which the reviewer found to be well-justified), but then later used a functional variant of PCA on the actual data themselves.

We should clarify that the functional PCA and UMAP are conducted on different data, and so are not directly comparable. We used UMAP on the centered and uncentered covariances matrices in order to illustrate the effects of our centering procedure. Functional PCA, by contrast, was applied to the excursion curves extracted from each task block as a way to summarize the main patterns of variability. We hope this is now clear.

Along the same lines, Reviewer 1 felt that the resting state is used as the baseline by which deviances are estimated. Wouldn't the Baseline period during the task be a more appropriate baseline for looking at learning related changes? It seems like using rest, without the same sensory or motor engagement, would be the wrong way to isolate excursions from the manifold from learning alone.

This is an insightful comment, and is something we considered carefully when planning the analysis. There is, of course, an argument to be made that using the baseline phase of the task as the manifold from which excursion is measured is more interpretable, and more directly assesses the deviations from manifold structure associated with the onset of rotation. However, it also introduces a key confound: As the end of the baseline block corresponds precisely with the onset of rotation, it is difficult to disentangle a jump in excursion due to (1) introduction of the visuomotor rotation itself versus (2) the jump expected due to the fact that we are now considering out-of-sample imaging volumes. Because PCA maximizes variance explained *in-sample*, we would expect larger excursions in out of sample observations, even if they were generated by the same process. That is, we would expect an increase in excursion during the learning block, even if network structure remained constant. Given this potential confound, we thus chose the resting state scan as the basis for our comparison. Note that this is also the reason for our centering with respect to resting state in the embedding analysis.

We have added text to the methods, essentially repeating the previous paragraph, in order to make our reasoning explicit for the reader.

…their subsequent assessment across the baseline/early/late stages allows for interpretations of only how the relative differences in the strength of (task)-components differs across learning compared to the resting state. Given the nature of the visuomotor rotation task, wouldn't the initial 'baseline' stage provide a more natural starting point to assess changes that occur with learning?

We agree with this comment as well. However, as the embedding is an attempt to characterize the changes in excursion, we wished to keep these analyses as consistent as possible. Since the excursion was computed relative to the subjects' resting state manifolds, we likewise chose to center subjects' covariance matrices with respect to their resting state scans.